# Mixture of Distributions Matters: Dynamic Sparse Attention for Efficient Video Diffusion Transformers

Yuxi Liu [1]  Yipeng Hu [1]  Zekun Zhang [2]  Kunze Jiang [3]
Kun Yuan [4]

## Abstract

While Diffusion Transformers (DiTs) have achieved notable progress in video generation, this long-sequence generation task remains constrained by the quadratic complexity inherent to self-attention mechanisms, creating significant barriers to practical deployment. Although sparse attention methods attempt to address this challenge, existing approaches either rely on over-simplified static patterns or require computationally expensive sampling operations to achieve dynamic sparsity, resulting in inaccurate pattern predictions and degraded generation quality. To overcome these limitations, we propose a **M**ixture-**O**f-**D**istribution **DiT** (**MOD-DiT**), a novel sampling-free dynamic attention framework that accurately models evolving attention patterns through a two-stage process. First, MOD-DiT leverages prior information from early denoising steps and adopts a distributed mixing approach to model an efficient linear approximation model, which is then used to predict mask patterns for a specific denoising interval. Second, an online block masking strategy dynamically applies these predicted masks while maintaining historical sparsity information, eliminating the need for repetitive sampling operations. Extensive evaluations demonstrate consistent acceleration and quality improvements across multiple benchmarks and model architectures, validating MOD-DiT's effectiveness for efficient, high-quality video generation while overcoming the computational limitations of traditional sparse attention approaches.

[1]Peking University [2]University of Electronic Science and Technology of China [3]University of Science and Technology of China [4]Center for Machine Learning Research, Peking University, Beijing, China. Correspondence to: Kun Yuan <Kunyuan@pku.edu.cn>.

*Proceedings of the 43rd International Conference on Machine Learning*, Seoul, South Korea. PMLR 306, 2026. Copyright 2026 by the author(s).

## 1. Introduction

The evolution from 2D+1D frameworks (Blattmann et al., 2023; Guo et al., 2023) to 3D full-attention Video Diffusion Transformers (vDiTs) (Peebles & Xie, 2023) has fundamentally advanced video synthesis, powering state-of-the-art models like CogVideoX (Yang et al., 2024), Hunyuan-Video (Kong et al., 2024), and Wan2.1 (Wang et al., 2025). However, the quadratic computational complexity inherent to 3D self-attention remains a critical bottleneck for practical deployment.

While recent sparse attention techniques attempt to mitigate this overhead (Chen et al., 2025; Xi et al., 2025; Li et al., 2025), they predominantly impose rigid, static sparsity patterns—such as fixed vertical grids or global exponential decay. These oversimplified constraints fail to capture the true dynamics of vDiTs, where attention distributions exhibit a continuously evolving, probabilistic mixture of structural patterns throughout the denoising process. This mismatch between static masks and fluid attention dynamics severely degrades generation quality. Consequently, bridging the gap between computational efficiency and high-fidelity video synthesis hinges on addressing two fundamental questions for vDiT inference:

> **Q1**. **How can we accurately model the dynamic mixture of distribution patterns in the attention maps of vDiTs?**

Current sparse attention methods rely on static, oversimplified representations of attention patterns. Sparse-vDiT (Chen et al., 2025) constrains patterns to vertical and block-diagonal structures, while SVG (Xi et al., 2025) categorizes attention heads into binary spatial and temporal types, and Radial Attention (Li et al., 2025) employs fixed global patterns based on energy decay. These approaches fail to capture a crucial characteristic: attention maps in vDiTs exhibit a dynamic mixture of three distinct patterns—block-diagonal for intra-frame temporal coherence, parallel-to-main-diagonal for inter-frame spatial correlation, and vertical for global dependencies across tokens. These patterns continuously evolve throughout the denoising process, making their accurate modeling essential for high-quality

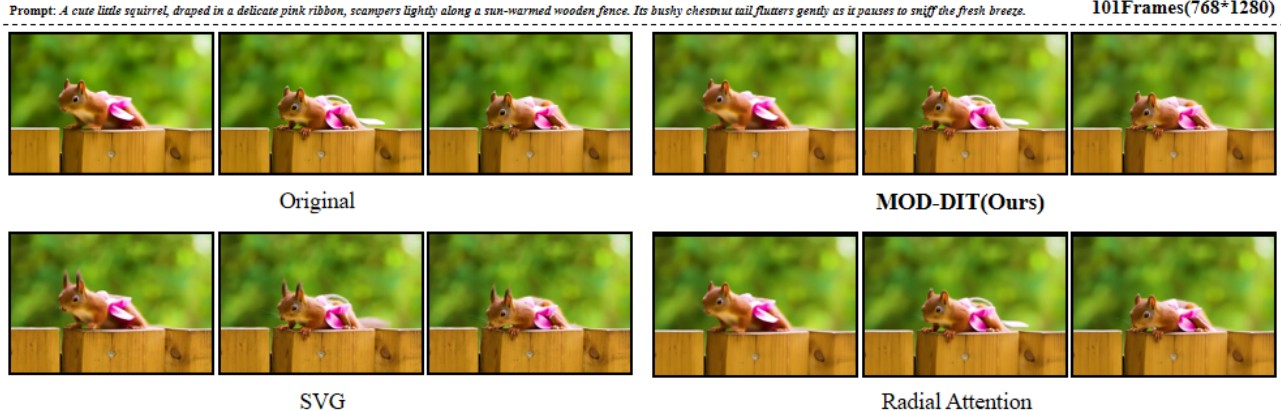

*Figure 1.* Comparison of the visualization effects of different sparse attention methods on HunyuanVideo(Kong et al., 2024). Our method MOD-DiT consistently achieves 2.05× speedup, and keep almost the same as original videos.

video generation.

> **Q2. How can we design a sampling-free dynamic sparse attention algorithm that adapts to denoising steps and enables high-performance implementation?**

Existing dynamic sparse attention techniques (Zhang et al., 2025b; Tan et al., 2025; Xia et al., 2025; Jiang et al., 2024) suffer from three critical limitations. First, sampling-based methods incur substantial computational overhead that negates their intended efficiency gains. Second, even purportedly dynamic approaches fail to adapt to evolving attention distributions—a rigidity that undermines the accuracy-efficiency trade-off. Third, these methods overlook the impact of dynamic denoising time steps on sparse attention, which is essential for capturing the evolving patterns in video diffusion transformers.

**Main Contributions.** To address the aforementioned open questions, this paper introduces **M**ixture-**O**f-**D**istribution **DiT** (**MOD-DiT**), a novel dynamic sparse attention framework for vDiTs. Our contributions are as follows:

- **Mixture Pattern identification.** We reveal that attention maps in vDiTs exhibit a dynamic mixture of three distinct distribution patterns—block-diagonal, parallel-to-main-diagonal, and vertical—that evolve throughout the denoising process. Grounded in rigorous theoretical insights and validated by extensive ablation studies, this discovery challenges existing static paradigms and establishes a mathematically sound foundation for accurate attention modeling. *(Addresses Q1)*

- **Training-free and sampling-free dynamic sparse attention.** We propose a plug-and-play algorithm that operates entirely online during inference, eliminating both training costs and sampling overhead. By leveraging prior information from early denoising steps to predict dynamic masks, our approach achieves fine-grained adaptivity, tailoring mask patterns and sparsity ratios simultaneously at three distinct levels: *input, head, and denoising step*. This tri-level dynamic routing rigorously optimizes the accuracy-efficiency trade-off and is hardware-optimized for seamless integration with mainstream APIs for scalable deployment. *(Addresses Q2)*

- **Superior performance.** Extensive experiments demonstrate that MOD-DiT achieves 1.75× on Wan2.1 (Wang et al., 2025) and 2.05× on hunyuan(Kong et al., 2024) model inference speedup while establishing excellent results on VBench (Huang et al., 2023). These results confirm the effectiveness of our solutions to both fundamental challenges. *(Validates solutions to Q1 and Q2)*

## 2. Related work

**Video Diffusion Models and Transformers.** Diffusion models have achieved state-of-the-art results in image synthesis (Ho et al., 2020), prompting their extension to video generation. Early approaches employed decoupled spatial and temporal attention (Blattmann et al., 2023), while recent advances adopt 3D dense attention (Melnik et al., 2024; Peebles & Xie, 2023) to jointly model spatio-temporal dynamics and capture long-range dependencies. Leading models including OpenSora (Zheng et al., 2024), CogVideoX (Yang et al., 2024), HunyuanVideo (Kong et al., 2024), and Wan2.1 (Wang et al., 2025) now support diverse applications spanning animation, editing, and long-video generation (Qi et al., 2023; Chen et al., 2024; Henschel et al., 2025; Hu et al., 2026). However, the quadratic complexity of self-attention remains a fundamental bottleneck.

**Sparse Attention for Efficient Video Diffusion Transformers.** Sparse attention methods address computational

costs by restricting token interactions through either static or dynamic designs. Static approaches employ fixed sparsity patterns: Sparse-vDiT (Chen et al., 2025) predefines vertical and block-diagonal patterns, Radial Attention (Li et al., 2025) introduces exponential decay achieving $\mathcal{O}(n \log n)$ complexity, while Sliding Tile Attention (Zhang et al., 2025c) utilize fixed 3D windows and distance-based restrictions. In contrast, dynamic methods adapt patterns based on input characteristics: Sparse VideoGen (Xi et al., 2025) classifies attention heads as spatial or temporal through online sampling, MInference (Jiang et al., 2024) explores hybrid static-dynamic patterns, AdaSpa (Xia et al., 2025) captures hierarchical sparsity via blockified dynamic patterns while achieving training-free acceleration with LSE-cached online search, SpargeAttention (Zhang et al., 2025b) emphasizes cross-modal compatibility with adaptive structures, and LiteAttention (Shmilovich et al., 2025) leverages temporal coherence across denoising steps to early identify and propagate computationally skipable blocks, reducing redundant computations through block-level skipping. Specifically, SpargeAttention primarily relies on query and key features, while AdaSpa extracts signals from attention block values—neither leverages the critical observation that attention maps inherently manifest as superpositions of structured patterns with gradual evolution. This oversight leads to suboptimal feature extraction efficiency and accuracy. Despite these advances, existing methods fail to capture the **true dynamic mixture of attention patterns** in vDiTs, motivating our approach.

**Efficient Diffusion and Attention Mechanisms.** Complementary efficiency techniques for diffusion models include pruning to reduce parameters (Ma et al., 2025), quantization to lower bit-width overhead (Zeng et al., 2025; Tian et al., 2024), and caching strategies that trade memory for speed (Qiu et al., 2025; Liu et al., 2025; Kahatapitiya et al., 2024a). Video-specific methods— TeaCache (Liu et al., 2025), FasterCache (Lv et al., 2025), and AdaCache (Kahatapitiya et al., 2024b)—exploit temporal redundancy by reusing features across adjacent denoising steps, while distillation approaches reduce inference timesteps (Park et al., 2024) and high-ratio VAEs compress latent representations. In broader attention research, sparse patterns have been extensively explored: vision models including Swin Transformer (Liu et al., 2021), NAT (Guo et al., 2020), and NLP models like Longformer (Beltagy et al., 2020) utilize sliding windows. Recent advances such as MInference (Jiang et al., 2024) and FlexPrefill (Lai et al., 2025) have explored diverse sparsity patterns for efficient attention computation, yet few are tailored to the unique spatiotemporal dynamics and denoising step dependencies of video diffusion transformers.

# 3. Preliminaries

**Full and Sparse Attention.** Let $B_b$ denote batch size, $N$ total tokens, $H$ the number of attention heads, and $d = D/H$ the dimension per head. Input features $I \in \mathbb{R}^{B_b \times N \times D}$ are linearly projected into query, key, and value tensors $Q, K, V \in \mathbb{R}^{B_b \times H \times N \times d}$. For each head $h$, standard full attention computes the output $O_h \in \mathbb{R}^{B_b \times N \times d}$ via an all-to-all interaction:

$$O_h = \text{softmax}\left(\frac{Q_h K_h^T}{\sqrt{d}}\right) V_h$$

However, this incurs an $\mathcal{O}(N^2)$ complexity that bottlenecks long-sequence video generation. Sparse attention mitigates this by restricting token interactions through a binary mask $M \in \mathbb{R}^{N \times N}$, where $M_{p,q} \in \{-\infty, 0\}$. Setting $M_{p,q} = -\infty$ drops the interaction between query token $p$ and key token $q$. Formally, sparse attention is defined as:

$$\text{SA}(Q, K, V) = \text{softmax}(A + M)V, \qquad (1)$$

where $A = QK^T/\sqrt{d}$ represents the scaled attention scores.

**Dynamic Sparse Attention.** Current dynamic methods (Xia et al., 2025; Jiang et al., 2024; Xi et al., 2025; Zhang et al., 2025b) adapt masks across inputs and attention heads, but critically **overlook the denoising step dimension**. This rigidity prevents them from capturing the evolving temporal dynamics of diffusion processes (see Figures 3 and 9).

**Token-Type-Aware Masking.** Modern vDiTs (e.g., CogVideoX, HunyuanVideo) typically adopt a *decoder-only* unified attention architecture where text and video tokens are concatenated. To guarantee robust cross-modal alignment with negligible overhead, MOD-DiT applies full attention to text tokens (typically $\leq 1\%$ of the sequence), exclusively restricting $M$ for video-video interactions.

# 4. Mixture of Patterns in Attention

### 4.1. Dynamic Evolution of Attention Sparsity Maps

**Attention Sparsity Map.** We process each attention head independently: for the attention map $A_h$ of the $h$-th head, we partition $A_h$ into non-overlapping blocks of size $B \times B$, yielding $n = N/B$ blocks per dimension, and convert it to a sparsity map $S \in \mathbb{R}^{n \times n}$. The sparsity value for block $(i, j)$ is defined as:

$$S_{h,(i,j)} = \frac{1}{B^2} \sum_{x=0}^{B-1} \sum_{y=0}^{B-1} \mathbb{I}\left(A_{h,(iB+x, jB+y)} < \eta\right), \quad (2)$$

where $\eta$ denotes the sparsity threshold and $\mathbb{I}(\cdot)$ is the indicator function. Less $S_{h,(i,j)}$ indicates more informative blocks. For the convenience of symbol reading, we uniformly use $S$ instead of $S_h$ in the subsequent descriptions and formulas.

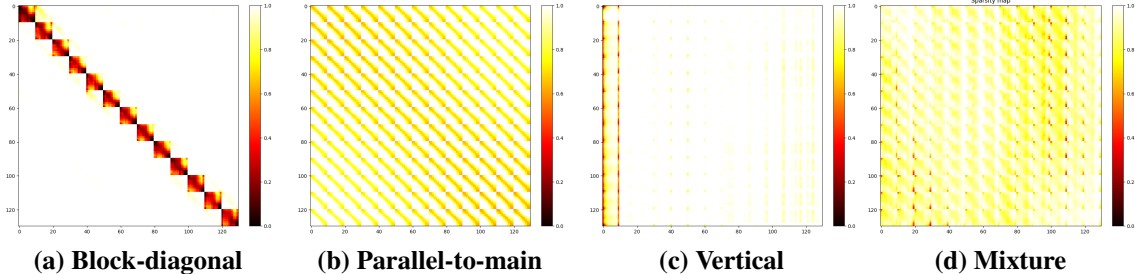

**(a) Block-diagonal**  **(b) Parallel-to-main**  **(c) Vertical**  **(d) Mixture**

*Figure 2.* Visualization of the four attention patterns in CogVideoX-v1.5(Yang et al., 2024).

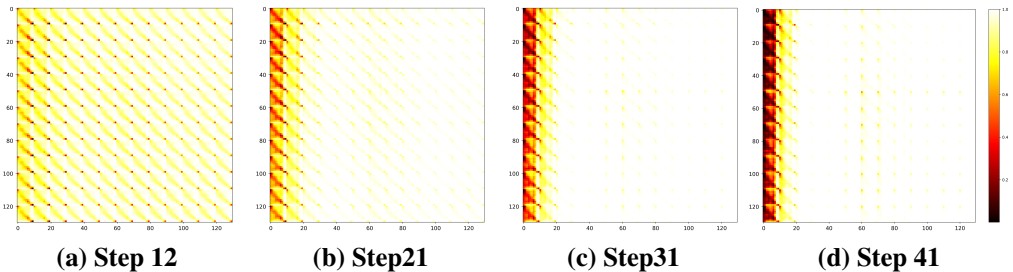

**(a) Step 12**  **(b) Step21**  **(c) Step31**  **(d) Step 41**

*Figure 3.* Evolution of the Sparsity Map in CogVideoX-v1.5(Yang et al., 2024) (Layer 0, Head 9) over Denoising Steps, demonstrating significant differences in the attention sparsity patterns across denoising steps.

**Pattern Mixtures.** As illustrated in Figure 2 and 3, the attention sparsity map $S^{(t)} \in \mathbb{R}^{n \times n}$ exhibits a fundamental structural evolution during the denoising process. It asymptotically converges into three distinct topological bases: block-diagonal for intra-frame spatial coherence, parallel-to-main-diagonal for inter-frame temporal continuity, and vertical for global cross-token dependencies. Particularly in mid-to-late denoising stages, the attention manifold manifests as a dynamic superposition of these three bases, as exemplified in pattern (d).

### 4.2. Generalized Linear Approximation Model

To accurately model the dynamic pattern mixture in the attention sparsity map $S^{(t)} \in \mathbb{R}^{n \times n}$ at denoising step $t$, we propose a linear approximation model that unifies three core structural priors:

$$S^{(t)} = \sum_{k=1}^{2n-1} c_k^{(t)} C_k + \sum_{k=1}^{n} d_k^{(t)} D_k + \sum_{k \in \mathcal{A}} e_k^{(t)} E_k + R^{(t)}, \quad (3)$$

where $n = N/B$ denotes the number of blocks per dimension ($N$ being the total number of tokens and $B$ the block size). The terms $\{C_k\}$, $\{D_k\}$, and $\{E_k\} \in \{0,1\}^{n \times n}$ are binary basis matrices representing the three core patterns:

- **Parallel-diagonal patterns** ($\{C_k\}_{k=1}^{2n-1}$): Model inter-frame spatial correlation, where $\delta_k = k - n + 1$ represents the diagonal offset.
- **Vertical patterns** ($\{D_k\}_{k=1}^{n}$): Capture global cross-token dependencies through column-wise structures.

- **Block-diagonal patterns** ($\{E_k\}_{k \in \mathcal{A}}$): Maintain intra-frame temporal coherence, where $\mathcal{A}$ is the set of valid block indices and each matrix activates a specific block region $B_k$.

Additionally, $c_k^{(t)}, d_k^{(t)}, e_k^{(t)} \in \mathbb{R}$ are intensity scalars quantifying the contribution of each respective pattern at step $t$, and $R^{(t)} \in \mathbb{R}^{n \times n}$ is the residual matrix capturing unstructured approximation errors.

To solve for the optimal coefficient vector $\mathbf{X}^{(t)} = [c_1^{(t)}, \ldots, e_{|\mathcal{A}|}^{(t)}]^T \in \mathbb{R}^{3n-1+|\mathcal{A}|}$, we reformulate Eq. (3) as a least-squares problem:

$$\mathbf{X}^{(t)} = \arg\min_{\mathbf{X}} \left\| \text{vec}(S^{(t)}) - M\mathbf{X} \right\|_2^2, \quad (4)$$

where $M = [\text{vec}(C_1), \ldots, \text{vec}(E_{|\mathcal{A}|})]$ is the design matrix.

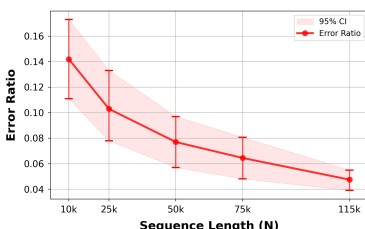

*Figure 4.* Normalized approximation error of linear approximation model (eq. (3)) across sequence lengths on hunyuan video, We sampled 200 sets of error data for each sequence length and plotted the average error and its 95% confidence interval (CI).

We validate this model across varying sequence lengths $N$ using the **normalized approximation error**: $\text{NAE}(N) = \|R^{(t)}\|_2 / \|S^{(t)}\|_2$. As illustrated in Figure 4, the NAE is not only consistently low but notably decreases as $N$ scales up. This asymptotic scaling confirms that Eq. (3) captures the core structural patterns of attention sparsity, proving inherently more accurate for longer sequences. A hardware-optimized kernel is implemented to guarantee real-time inference (see Appendix C).

*Remark* 4.1. The least-squares kernel in the Appendix C accelerates solving by leveraging the sparsity and special structure of basis matrices to **decompose high-complexity matrix operations into low-dimensional structured computations**, combined with GPU multi-level parallelism to optimize memory access and efficiency.

### 4.3. Pattern Intensity Evolution

Using the linear model (Eq. (4)), we analyze the evolution of attention sparsity maps $S^{(t)}$ by solving for the intensity scalars of the vertical and parallel-diagonal patterns. Since the block-diagonal pattern remains largely invariant, we characterize it via static thresholding (Section 5.3) rather than dynamic prediction. As Figure 5 illustrates, while early denoising steps exhibit complex non-linear intensity fluctuations, they **converge to stable piecewise linear trends in mid-to-late stages**.

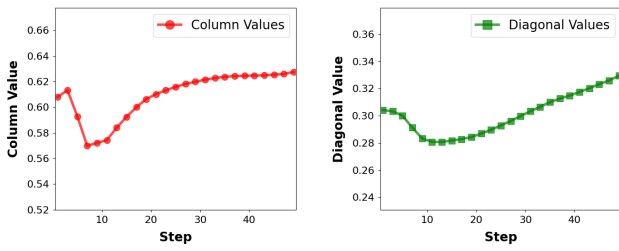

*Figure 5.* Evolution of vertical and parallel-diagonal pattern intensities across denoising steps, showing convergence to piecewise linearity.

This piecewise linearity enables predictive masking: by modeling $c_k^{(t)}$ and $d_k^{(t)}$ as linear functions of $t$, we design a pattern predictor (Section 5.2) that dynamically adjusts masks across denoising steps.

**Universality across prompts.** A large-scale statistical validation across diverse prompts, layers, and heads (Appendix A.1.11) confirms that the Normalized reconstruction error (NRE) of this piecewise linearity consistently falls within $[0.00, 0.10]$ (Figure 14). This demonstrates that while the absolute evolutionary slopes depend on specific video content, the *structural stability* in the mid-to-late regime is a universal property, providing a robust foundation for our prediction mechanism.

### 4.4. Theoretical Insights: Inevitability and Sufficiency

**Diffusion Convergence Guarantees Predictability.** Studies on diffusion dynamics confirm that during mid-to-late denoising steps, attention distributions transition from chaotic high-frequency noise to stable, semantically structured patterns (Tumanyan et al., 2022; Hertz et al., 2022). This structural collapse ensures that the **attention manifold becomes highly predictable** and can be spanned by a fixed set of basis distributions.

**The Inevitability of the Three Basis Patterns.** Under the universal frame-first token layout, let $p = (t_p, s_p)$ and $q = (t_q, s_q)$ denote the spatiotemporal coordinates of the query and key tokens, respectively, where $t$ represents the temporal index and $s$ the spatial coordinates. The high-energy attention manifold can be strictly decoupled into three orthogonal interaction subspaces:

- **Block-diagonal ($\mathcal{P}_{\text{spatial}}$):** Governs intra-frame spatial coherence, formally defined by the subspace $\{(p, q) \mid t_p = t_q\}$. It captures dense spatial interactions within identical temporal frames.
- **Parallel-to-main-diagonal ($\mathcal{P}_{\text{temporal}}$):** Maintains inter-frame temporal continuity, defined by $\{(p, q) \mid t_p \neq t_q, \|s_p - s_q\| \to 0\}$. It models the temporal progression of spatially aligned or locally shifted regions across different frames.
- **Vertical global ($\mathcal{P}_{\text{global}}$):** Ensures cross-frame semantic alignment, defined by $\{(p, q) \mid q \in \Omega_{\text{anchor}}\}$. Here, $\Omega_{\text{anchor}}$ represents a concentrated subset of globally dominant key tokens (e.g., semantic anchors or attention sinks) that persistently broadcast to all queries.

*Remark* 4.2. The union of these three independent subspaces, $\mathcal{S} = \mathcal{P}_{\text{spatial}} \cup \mathcal{P}_{\text{temporal}} \cup \mathcal{P}_{\text{global}}$, establishes a theoretically complete basis for the **principal high-interaction manifold** in video generation. This sufficiency is dictated by the information bottleneck of video dynamics: any essential high-energy attention must physically correspond to intra-frame spatial composition, inter-frame temporal tracking, or cross-frame semantic anchoring. Interactions falling outside this tri-basis union represent **non-causal spatiotemporal jumps**, which, in mid-to-late denoising stages, mathematically degenerate into negligible **low-rank background noise**. This physical intuition is definitively corroborated by our quantitative validations, where the least-squares projection onto this basis consistently achieves an average Normalized Approximation Error (NAE) of $< 0.14$. By effectively capturing over $90\%$ of the total attention energy, it confirms that no critical interaction dimensions are omitted, rendering this linear combination a rigorously sufficient approximation.

# 5. Method

We propose the **MOD-DiT** framework (detailed in Algorithm 1), built upon the empirical insights from Section 4. It achieves sampling-free, dynamic sparse attention through a streamlined three-step pipeline: sparsity map reconstruction, linear pattern prediction, and dynamic mask generation.

**Key Notations.** We define: $t$ as the denoising step, $t_p^{(i)} = m + i \cdot \Delta t$ $(i \geq 0)$ as the linear prediction step with interval $\Delta t$, and $S^{(t)} \in \mathbb{R}^{n \times n}$ as the attention sparsity map.

**Warm-Up with Full Attention.** As shown in Figure 5, in the early denoising stage, the sparsity map $S^{(t)}$ exhibits **unstructured spatiotemporal dependencies**, making direct application of sparse attention via Eq.(1) prone to irreversible information loss. To address this, we perform warm-up via $m$ steps of full attention. Our sensitivity analysis confirms that $m$ is a tunable hyperparameter; reducing $m$ to 8 steps further enhances speed with marginal quality impact, offering a flexible trade-off for various inference budgets. See ablation 10 for more details.

## 5.1. Sparsity Map Reconstruction

Accurately estimating the pattern intensities via our linear approximation model (Eq.(3)) necessitates a dense attention matrix. Because sparse inference inherently computes only a masked subset of attention scores $A_{\text{masked}}^{(t_p^{(i)})}$, direct pattern extraction becomes mathematically ill-posed. To bridge this gap, we propose an iterative temporal fusion strategy to reconstruct a proxy dense attention map. This strategy synthesizes the currently computed masked map with the historical reconstructed map:

$$\hat{A}^{(t_p^{(i+1)})}(p,q) = \begin{cases} A_{\text{masked}}^{(t_p^{(i+1)})}(p,q), & M^{(t_p^{(i+1)})}(p,q) = 0 \\ \hat{A}^{(t_p^{(i)})}(p,q), & M^{(t_p^{(i+1)})}(p,q) = -\infty \end{cases}$$
(5)

Here, $\hat{A}^{(t_p^{(0)})} = A^{(m)}$ initializes the sequence using the dense map from the final full-attention warm-up step ($t = m$). Periodically updating this reconstruction every $\Delta t$ steps acts as a structural anchor, effectively mitigating the cumulative drift and bias introduced by continuous linear prediction. Finally, this proxy dense map $\hat{A}^{(t)}$ is converted into the sparsity map $\hat{S}^{(t)}$ (Eq.(2)) to estimate the latest pattern intensities, ensuring seamless integration with our linear model. Empirical ablations confirm that this fusion strategy introduces negligible approximation error (see Appendix A).

## 5.2. Linear Prediction of Vertical & Parallel Pattern

As observed in Figure 5, the intensity scalars of parallel-diagonal ($c_k^{(t)}$) and vertical-line ($d_k^{(t)}$) patterns in attention sparsity maps exhibit stable piecewise linearity during the

mid-late denoising stage (i.e., $t > m$), a property validated by our systematic ablation study with 300 data points (Appendix A.1.11). Leveraging this regularity, we design a lightweight linear prediction method to generate sparsity maps for subsequent denoising steps.

**Prediction Principle and Formulation.** For any three consecutive prediction steps $t_p^{(i)}$, $t_p^{(i+1)}$, and $t_p^{(i+2)}$, we use the reconstructed sparsity maps $\hat{S}^{(t_p^{(i)})}$ and $\hat{S}^{(t_p^{(i+1)})}$ to linearly predict the intensity scalars $\hat{c}_k^{(t)}$ and $\hat{d}_k^{(t)}$ for all denoising steps $t \in (t_p^{(i+1)}, t_p^{(i+2)}]$.

$$\hat{c}_k^{(t)} = \hat{c}_k^{(t_p^{(i+1)})} + \frac{\hat{c}_k^{(t_p^{(i+1)})} - \hat{c}_k^{(t_p^{(i)})}}{t_p^{(i+1)} - t_p^{(i)}} \cdot \left(t - t_p^{(i+1)}\right)$$

$$\hat{d}_k^{(t)} = \hat{d}_k^{(t_p^{(i+1)})} + \frac{\hat{d}_k^{(t_p^{(i+1)})} - \hat{d}_k^{(t_p^{(i)})}}{t_p^{(i+1)} - t_p^{(i)}} \cdot \left(t - t_p^{(i+1)}\right)$$

where $\hat{c}_k^{(t_p^{(i)})}$, $\hat{d}_k^{(t_p^{(i)})}$ are extracted from $\hat{S}^{(t_p^{(i)})}$; $\hat{c}_k^{(t_p^{(i+1)})}$, $\hat{d}_k^{(t_p^{(i+1)})}$ are derived from $\hat{S}^{(t_p^{(i+1)})}$ by eq.4.

This piecewise linear prediction is critical for quality preservation. Ablation shows static one-time prediction degrades performance significantly, confirming the necessity of dynamic tracking (see Appendix A for details).

> *Remark* 5.1. **Robustness via Predict-Correct Pipeline.** Attention dynamics vary across video content. To prevent error accumulation from long-term extrapolation, MOD-DiT employs a *short-cycle predict-correct pipeline* (Sec 5.1 and 5.2). By exclusively activating linear prediction in the stable mid-to-late regime ($t > m$), early highly non-linear dynamics are safely handled by the full-attention warm-up. Crucially, we perform local extrapolation over short intervals ($\Delta t$) and periodically reconstruct the global sparsity map (Eq. 5) to re-anchor predictions, ensuring dynamic adaptation to prompt-specific attention shifts.

## 5.3. Dynamic Mask Generation

We design a dynamic mask generation strategy that integrates pattern-based Top-K selection and conditional block-diagonal preservation. For each attention head $h$ and denoising step $t$, the mask $M_h^{(t)}(p,q)$ is defined as

$$M_h^{(t)}(p,q) = \begin{cases} 0, & (p,q) \in \mathcal{K}_t \cup \mathcal{K}_E \\ -\infty, & \text{otherwise} \end{cases}$$

where $\mathcal{K}_t$ denotes the Top-$K$ dynamic pattern set, constructed by ranking the predicted intensity scalars $\{\hat{c}_k^{(t)}\} \cup \{\hat{d}_k^{(t)}\}$ of the parallel-diagonal ($C_k$) and vertical ($D_k$) patterns. Meanwhile, $\mathcal{K}_E = \text{supp}(E)$ represents the

block-diagonal support, conditionally retained if $e_{\min} = \min\{\hat{e}^{(m-1)}, \hat{e}^{(m)}\} > \tau_e$. Notably, because the block-diagonal pattern encodes structurally invariant intra-frame spatial interactions across denoising steps, its presence can be efficiently verified via this static thresholding. This elegantly bypasses the continuous linear prediction required by the dynamically evolving inter-frame and global patterns.

The final layer-wise mask $M^{(t)}$ is constructed by concatenating masks across all $H$ attention heads

$$M^{(t)} = \text{cat}\left(M_1^{(t)}, M_2^{(t)}, \ldots, M_H^{(t)}\right).$$

### 5.4. Hardware Acceleration

To fully leverage the efficiency advantages of MOD-DiT, we designed three optimization strategies.

**Optimized Least-Squares Kernel.** We designed an optimized least-squares kernel based on CUDA, reducing the solving time of E.q (4) by $100\times$ compared to native solvers 'torch.lstsq' in PyTorch. See Appendix B for more details.

**Hybrid Attention Execution.** We use FlashAttention-2(Dao, 2023) in the warm-up stage to minimize the computational overhead of full attention; in the sparse stage, we switch to SageAttention (Zhang et al., 2025a), which directly skips invalid token interactions at the hardware level, further reducing redundant computations and improving execution speed.

**Block-wise Attention Computation.** We adopt a block-wise strategy for attention calculation acceleration(Jiang et al., 2024): we partition query (Q) and key (K) tensors into non-overlapping blocks with a fixed block size of 128 via e.q (2). This block-wise execution enables localized memory access and leverages GPU warp-level parallelism to process intra-block token interactions in batches, further lowering the latency of attention computation.

*Remark* 5.2. The computational overhead introduced by **MOD-DiT** is negligible, accounting for merely 1–2% of the full attention latency as empirically measured on VBench (Huang et al., 2023). Detailed theoretical and empirical profiling is provided in Appendix C.

## 6. Experiments

### 6.1. Experimental Settings

**Models.** We evaluate MOD-DiT both on popular small and large video generation models including CogVideoX-v1.5-2b(Yang et al., 2024), HunyuanVideo-13b(Kong et al., 2024) and Wan2.1-14b(Wang et al., 2025).

**Main Baselines.** We compare MOD-DiT with latest sparse attention methods:

- **SVG(Xi et al., 2025)**: Accelerates video models with sparse attention by dynamically classifying attention heads as spatial or temporal and applying corresponding masks.
- **Radial(Li et al., 2025)**: Accelerates video models by Constructing a static mask with Spatiotemporal Energy Decay in video generation.
- **MInference(Jiang et al., 2024)**: A classical sparse acceleration technique migrated from large language models. which choose different sparse patterns for different heads.
- **SpargeAttn(Zhang et al., 2025b)**: Accelerates attention calculation by pruning blocks and Online softmax threshold filtering.
- **LiteAttention(Shmilovich et al., 2025)**: An efficient sparse attention method that leverages temporal coherence between denoising steps by early propagating computationally skipable blocks.

**Datasets.** We use the VBench(Huang et al., 2023) dataset to rate MOD-DiT and other baselines, which demonstrate the quality of generated videos across multiple dimensions.

**Metrics.** We evaluate generated videos using VBench (Huang et al., 2023), specifically focusing on the Imaging Quality and Subject Consistency dimensions. To assess visual fidelity and perceptual deviation from the original full-attention models, we report Peak Signal-to-Noise Ratio (PSNR), Structural Similarity Index Measure (SSIM), and Learned Perceptual Image Patch Similarity (LPIPS). Finally, **sparsity** is defined as the ratio of computed $B \times B$ non-overlapping blocks to the total block count.

### 6.2. Experimental Results Analysis

**Validation of the Linear Manifold Hypothesis.** The consistently low NAE shown in Figure 4 and the superior Sparsity Mask Fidelity Score (SMFS) in Figure 11 empirically validate our theoretical logic chain. This demonstrates that MOD-DiT captures the **intrinsic geometric skeleton** of the attention mechanism, which is the primary driver behind our 2.05x speedup without quality loss.

**Impact of Sequence Length.** We evaluated the impact of sequence length on inference time using the Hunyuan model (Kong et al., 2024), adjusting Top-$K$ hyperparameter of MOD-DiT accordingly. As shown in Figure 6, the acceleration achieved by MOD-DiT becomes more substantial than Full Attention as the sequence length increases.

**Quality Evaluation.** We evaluate MOD-DiT against baselines on mainstream video generation models under strictly controlled settings in table 1(12 full-attention warm-up steps, $\Delta t = 10$). MOD-DiT achieves an optimal sparsity-quality balance, delivering maximum inference speedups while maintaining highly competitive performance across core quantitative and qualitative metrics, which validate its effectiveness in optimizing the quality-efficiency tradeoff.

*Table 1.* Quantitative comparison on HunyuanVideo (Kong et al., 2024), and Wan2.1 (Wang et al., 2025) using VBench prompts. To ensure strict fairness, all sparse baselines utilize the same inference API. Model configurations: HunyuanVideo (13B, 117frames, 768 × 1280, A100 GPU), and Wan2.1 (14B, 69 frames, 768 × 1280, A100 GPU). Our method consistently achieves the best quality-efficiency trade-off, delivering maximum speedups and top scores while maintaining the **highest sparsity**.

| Model | Method | Sparsity | Quality | | | | | | | | | | Efficiency | |
| --- | --- | --- | --- | --- | --- | --- | --- | --- | --- | --- | --- | --- | --- | --- |
| | | | PSNR↑ | SSIM↑ | LPIPS↓ | S.C.↑ | B.C.↑ | M.S.↑ | T.F.↑ | I.Q.↑ | A.Q.↑ | D.D.↑ | Latency(s)↓ | Speedup↑ |
| HunyuanVideo | Full | 0.00% | - | - | - | 0.9582 | 0.9478 | 0.9766 | 0.9723 | 0.6693 | 0.6381 | 0.9215 | 6978s | 1.00× |
| | MInference | 67.1% | 18.21 | 0.638 | 0.490 | 0.9300 | 0.9180 | 0.9580 | 0.9500 | 0.6450 | 0.5650 | 0.9080 | 5286s | 1.32× |
| | Radial | 75.55% | 26.72 | 0.885 | 0.125 | 0.9312 | 0.9290 | 0.9715 | 0.9322 | 0.6467 | 0.5715 | 0.8423 | 3731s | 1.87× |
| | SVG | 71.22% | 26.44 | 0.861 | 0.170 | 0.8301 | 0.8711 | 0.9544 | 0.9018 | 0.5928 | 0.5330 | 0.9011 | 3834s | 1.82× |
| | Sparge | 68.00% | 25.43 | 0.842 | 0.195 | 0.9339 | 0.9156 | 0.9528 | 0.9510 | 0.6432 | 0.5603 | 0.9120 | 4105s | 1.70× |
| | **Ours** | **83.23%** | **27.73** | **0.879** | **0.119** | 0.9398 | 0.9320 | 0.9687 | 0.9527 | 0.6587 | 0.5899 | 0.9104 | **3405s** | **2.05×** |
| Wan2.1 | Full | 0.00% | - | - | - | 0.9623 | 0.9655 | 0.9844 | 0.9789 | 0.6722 | 0.6012 | 0.9378 | 3375s | 1.00× |
| | MInference | 60.5% | 15.81 | 0.675 | 0.343 | 0.9100 | 0.9200 | 0.9750 | 0.9650 | 0.6550 | 0.5700 | 0.9150 | 2557s | 1.32× |
| | Radial | 71.33% | 21.57 | 0.818 | 0.167 | 0.9152 | 0.9339 | 0.9860 | 0.9750 | 0.6620 | 0.5783 | 0.8500 | 2021s | 1.67× |
| | SVG | 69.08% | 20.93 | 0.795 | 0.222 | 0.7986 | 0.8859 | 0.9556 | 0.9347 | 0.6133 | 0.5292 | 0.9167 | 2109s | 1.60× |
| | Sparge | 50.10% | 18.67 | 0.735 | 0.198 | 0.9033 | 0.9275 | 0.9798 | 0.9632 | 0.6645 | 0.5610 | 0.9043 | 2296s | 1.47× |
| | **Ours** | **81.37%** | **22.75** | **0.821** | **0.152** | 0.9427 | 0.9488 | 0.9823 | 0.9752 | 0.6674 | 0.5827 | 0.9289 | **1929s** | **1.75×** |

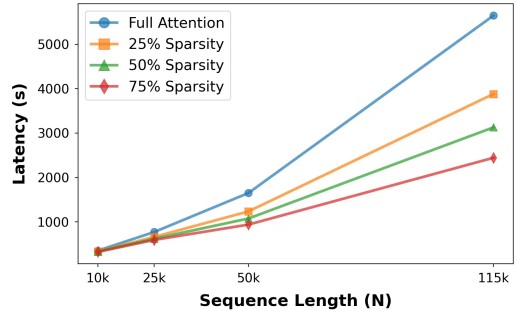

*Figure 6.* Comparison of inference time between Full Attention and MOD-DiT with various sparsity settings under varying sequence lengths.

To further substantiate these results, we detail several extended evaluations in Appendix A:

- **CogvdeioX-v1.5 results.** Due to space constraints, we include the full quantitative evaluation results on CogVideoX-v1.5 in Appendix 5.
- **Comparison with SVG-2.** We conduct a direct, head-to-head comparison with the latest SVG-2 (Yang et al., 2025) baseline, demonstrating MOD-DiT's sustained performance lead (See table 6).
- **Step-Distilled Generalization.** We verify MOD-DiT's robust compatibility and acceleration capabilities when applied to highly optimized, step-distilled diffusion models (See Sec A.4 ).

**Fidelity of Pattern Modeling.** As illustrated in Figure 13, the sparsity masks generated by MOD-DiT faithfully preserve the structural characteristics of the ground-truth attention maps, significantly outperforming baseline methods. This high visual fidelity confirms that MOD-DiT successfully **captures the dynamic, blended attention patterns**

**inherent** to video diffusion transformers.

**Ablation Study.** We also conduct ablation and validation experiments, including those on top-k selection, sparsity construction threshold $\eta$, reconstruction interval $\Delta t$, reconstruction Error, warm up steps $m$, sparsity map dynamic deviation, linearity of pattern intensity scalars, block size, and fairness verification under matched sparsity. All details are presented in the Appendix A.

# 7. Conclusion and limitation

**Conclusions.** MOD-DiT is a training-free dynamic sparse attention framework that models the evolution of three core attention patterns via sampling-free linear prediction. It seamlessly accelerates vDiTs without retraining, maintaining high video quality. Future work will integrate model quantization and feature caching for greater efficiency.

**Limitations.** The full-attention warm-up phase captures initial unstructured dependencies but may introduce computational overhead in short-sequence scenarios.

# Acknowledgments

This work is funded by the National Natural Science Foundation of China (No. 92370121, 12301392, 12288101, W2441021), and the National Key Research and Development Program of China (No. 2024YFA1012902). This research is also supported by the AI for Science Institute, Beijing, China and the National Engineering Labratory for Big Data Analytics and Applications.

## Impact Statement

This paper presents work whose goal is to advance the field of Machine Learning, with a focus on understanding the dynamic attention patterns in video diffusion transformers and optimizing sparse attention for efficient inference. There are many potential societal consequences of our work, none which we feel must be specifically highlighted here.

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

# A. Additional Experiments

## A.1. Ablation Study

### A.1.1. ABLATION STUDY ON THE TOP $K$ HYPERPARAMETER

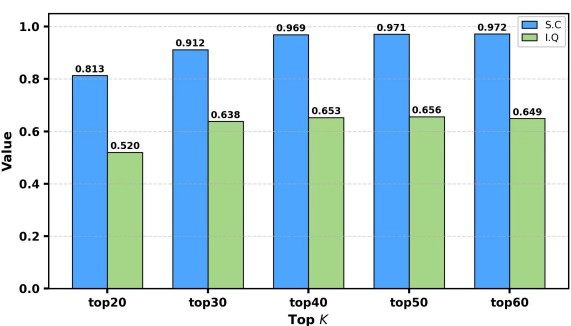

*Figure 7.* Ablation study on the Top $K$ hyperparameter.

Figure 7 shows the Top $K$ ablation on the Hunyuan(Kong et al., 2024) model. subject Consistency (S.C.) and Image Quality (I.Q.) initially increase with $K$ and then stabilize, indicating its performance sensitivity.

### A.1.2. ABLATION STUDY ON SPARSITY THRESHOLD $\eta$

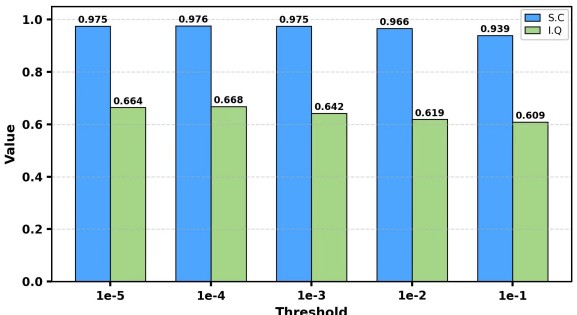

*Figure 8.* Ablation study on sparsity threshold $\eta$ in Eq.(2).

We evaluated the performance of MOD-DiT by adjusting the sparsity calculation threshold $\eta$. As shown in Figure 8, the results indicate that the threshold has a negligible impact on performance, with only a slight degradation occurring when the threshold is set to an extremely large value. Based on these findings, we selected a threshold of $1e-4$ for MOD-DiT.

### A.1.3. ABLATION STUDY ON DYNAMIC EVOLUTION OF ATTENTION SPARSITY MAPS

To further validate the dynamic evolution of attention sparsity maps and justify the necessity of modeling time-varying patterns (addressed in Q1), we conduct an additional experiment to quantify the structural deviation of sparsity maps across denoising steps. Specifically, we fix the warm-up steps to $m = 12$, and define the **difference error ratio** between the sparsity map at step $t \geq 12$ (post warm-up) and the map at the end of warm-up (step 12) as

$$\text{DER}(t) = \frac{\|S^{(t)} - S^{(12)}\|_2}{\|S^{(12)}\|_2}$$

where $S^{(t)}$ denotes the attention sparsity map at denoising step $t$. This metric quantifies how much the sparsity map deviates from its warm-up-converged state—higher values indicate more significant structural changes. As observed in Figure 9, the difference error ratio of most heads exhibits a certain degree of fluctuation across denoising steps. This confirms that the attention sparsity map undergoes substantial structural evolution even after the warm-up phase, rather than remaining static.

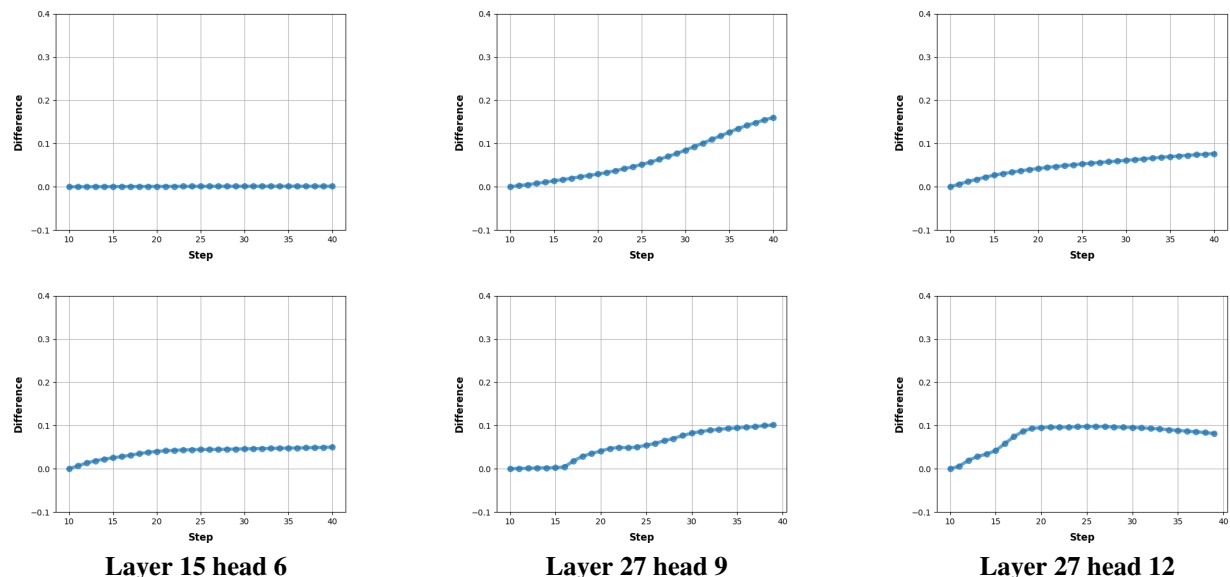

**Layer 15 head 6**                **Layer 27 head 9**                **Layer 27 head 12**

*Figure 9.* Normalized reconstruction error (NRE) results for 6 randomly sampled attention heads across varying denoising steps. Experiments are conducted on CogVideoX-v1.5 with 49-frame 640×512 videos on NVIDIA A100 80G GPUs.TBD

### A.1.4. ABLATION STUDY ON RECONSTRUCTION INTERVAL

To verify the robustness of MOD-DiT to the reconstruction interval $\Delta t$(defined in Section 5.1 ) and the effectiveness of our segment-wise prediction, we conduct ablation experiments with varying $\Delta t(1, 2, 3, 5, 10)$ and an additional "None" case—where only one pattern prediction is performed post-warm-up without subsequent mask updates. Experiments use evaluation metrics including Subject Consistency (SubConsist) and Imaging Quality (ImageQual).

*Table 2.* Ablation results of reconstruction interval $\Delta t$ on CogVideoX-v1.5. Experiments are conducted with 89-frame videos at 640×512 resolution using VBench prompts, on NVIDIA A100 80G GPUs.

| $\Delta t$ | **SubConsis** | **ImageQual** |
|---|---|---|
| 1 | 0.925 | 0.623 |
| 2 | 0.926 | 0.622 |
| 3 | 0.920 | 0.621 |
| 5 | 0.926 | 0.623 |
| 10 | 0.920 | 0.610 |
| None | 0.90 | 0.57 |

As observed in table 2, all tested $\Delta t$ values yield nearly identical performance, which confirms that MOD-DiT is robust to the reconstruction interval. In contrast, the "None" case shows a significant performance drop, validating the necessity of segment-wise prediction. By periodically reconstructing sparsity maps and updating masks, MOD-DiT dynamically tracks evolving attention patterns, ensuring high spatio-temporal coherence and visual quality.

### A.1.5. ABLATION STUDY ON SPARSITY

To eliminate the confounding effect of sparsity differences and conduct a strictly fair comparison, we evaluate MOD-DiT under the same sparsity ratio as Radial Attention on the Wan2.1 model. All experiments follow the unified setting: NVIDIA A100 80GB GPU, 69 frames, and 768×1280 resolution.

As shown in the results, under identical sparsity (i.e., the same computational cost), MOD-DiT achieves significantly better performance across all VBench dimensions compared to Radial Attention, with only a negligible increase in latency. This confirms that our dynamic mask design retains more critical attention information at the same computational overhead, leading to superior video generation quality.

*Table 3.* Results under matched sparsity settings with prior methods (Wan2.1 14B model, A100 80GB GPU, 69 frames, 768×1280 resolution)

| Method | S.C. | B.C. | M.S. | T.F. | I.Q. | A.Q. | D.D. | latency |
|---|---|---|---|---|---|---|---|---|
| Radial | 0.9152 | 0.9339 | 0.9860 | 0.9750 | 0.6620 | 0.5783 | 0.8500 | 1224s |
| **Ours** | 0.9518 | 0.9594 | 0.9867 | 0.9776 | 0.6630 | 0.6070 | 0.9789 | 1265s |

### A.1.6. ABLATION STUDY OF BLOCK SIZE

The block size is a key hyperparameter in MOD-DiT, directly affecting both the accuracy of sparsity mapping and computational efficiency. To investigate its impact on the quality-latency trade-off, we conduct an ablation study on the HunyuanVideo model under strictly aligned sparsity conditions. All experiments are performed on NVIDIA A100 80GB GPUs, with 117 frames and 768×1280 resolution.

*Table 4.* Ablation study of block size on model performance (Hunyuan model, A100 80GB GPU, 117 frames, 768×1280 resolution, aligned sparsity)

| Blocksize | S.C. | B.C. | M.S. | T.F. | I.Q. | A.Q. | D.D. | latency |
|---|---|---|---|---|---|---|---|---|
| 64 | 0.9401 | 0.9338 | 0.9689 | 0.9588 | 0.6451 | 0.5971 | 0.9152 | 1461s |
| 128 | 0.9398 | 0.9320 | 0.9687 | 0.9527 | 0.6587 | 0.5899 | 0.9104 | 1044s |

As shown in the results, increasing the block size from 64 to 128 brings a reduction in latency with negligible impact on generation quality. The performance across all VBench dimensions remains nearly identical, with some metrics (e.g., imaging quality) even slightly improved. This demonstrates that 128 is the optimal choice for balancing efficiency and accuracy, which we adopt as the default block size in our framework.

### A.1.7. ABLATION STUDY ON WARM-UP STEPS $m$

To determine the optimal number of warm-up steps for MOD-DiT, we conduct an ablation experiment focusing on the impact of $m$ on video generation quality. The warm-up phase is critical for capturing initial unstructured spatiotemporal dependencies (Section 5), and an appropriate $m$ ensures sufficient information retention without unnecessary computational overhead.

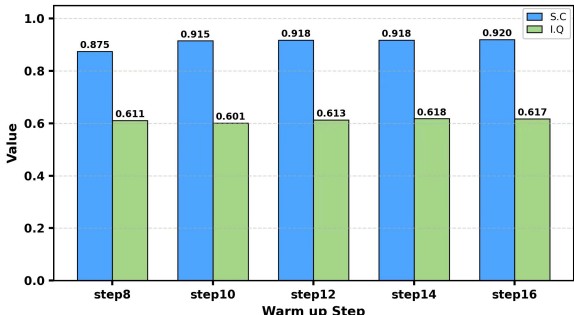

*Figure 10.* Ablation results of warm-up steps $m$ on CogVideoX-V1.5. Experiments are conducted with 89-frame videos at 640×512 resolution using VBench prompts, on NVIDIA A100 80G GPUs. We use Subject Consistency (SubConsist) and Imaging Quality (ImageQual) as evaluation metrics.

As observed in Figure 10 , excessive warm-up steps introduce redundant computational overhead without meaningful quality improvements. When m=12, MOD-DiT achieves near-optimal S.C. and I.Q. while avoiding unnecessary full-attention costs. Thus, we select 12 warm-up steps as the default setting.

### A.1.8. SUPPLEMENTARY QUANTITATIVE VALIDATION: SPARSITY MASK FIDELITY SCORE (SMFS)

To quantitatively evaluate the precision of our sparsity mask in capturing critical attention information, we propose the **Sparsity Mask Fidelity Score (SMFS)**. SMFS is defined as the ratio of the Frobenius norm of the masked attention map to

the norm of the original full attention map, which measures the information retention rate of the sparse mask:

$$\text{SMFS} = \frac{\|\mathbf{A} \odot \mathbf{M}\|_F}{\|\mathbf{A}\|_F}$$

where $\mathbf{A}$ denotes the original attention matrix, $\mathbf{M}$ is the binary sparsity mask, and $\odot$ represents element-wise multiplication. A lower SMFS value indicates that the sparse mask retains more critical attention information with fewer computations.

We conduct experiments on the HunyuanVideo model across various sequence lengths (10k, 25k, 50k, 75k tokens). We use 5 different prompts from VBench, randomly select 6 layers and 8 attention heads per prompt to calculate the average SMFS for each method.

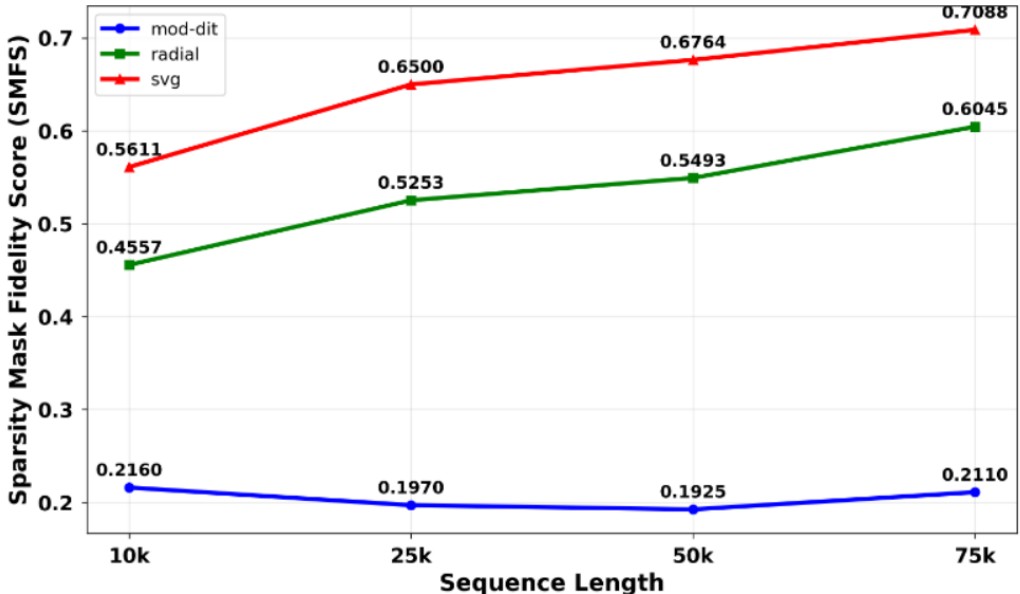

*Figure 11.* SMFS Comparison of Different Methods on HunyuanVideo Model. MOD-DiT consistently achieves significantly lower SMFS than SVG and Radial across all tested sequence lengths, demonstrating superior attention information capture accuracy.

As shown in Figure 11, MOD-DiT achieves significantly lower SMFS values than baseline methods (SVG and Radial Attention) at all sequence lengths. This confirms that our dynamic sparsity mask accurately captures the core structure of attention maps even at ultra-high sparsity ratios, which directly translates to better generation quality compared to static pattern-based baselines.

### A.1.9. ABLATION STUDY ON RECONSTRUCTION ERROR

To verify that the sparsity map reconstruction (Section 5.1) does not introduce significant errors that degrade pattern prediction accuracy, we analyze the difference between the reconstructed sparsity map from MOD-DiT and the ground-truth sparsity map from full attention across layers, attention heads, and denoising steps. We define the **normalized reconstruction error** (NRE) at step $t$ as

$$NRE(t) = \frac{\|\hat{S}^{(t)} - S_{GT}^{(t)}\|_2}{\|S_{GT}^{(t)}\|_2}$$

where $S_{GT}^{(t)}$ and $\hat{S}^{(t)}$ denotes the ground-truth sparsity map and the sparsity map reconstructed by MOD-DiT at denoising step $t$ respectively.

As shown in Figure 12, all sampled attention heads exhibit a stable and low normalized reconstruction error across denoising steps. This directly confirms that MOD-DiT introduces minimal errors when reconstructing attention maps via E.q (5).

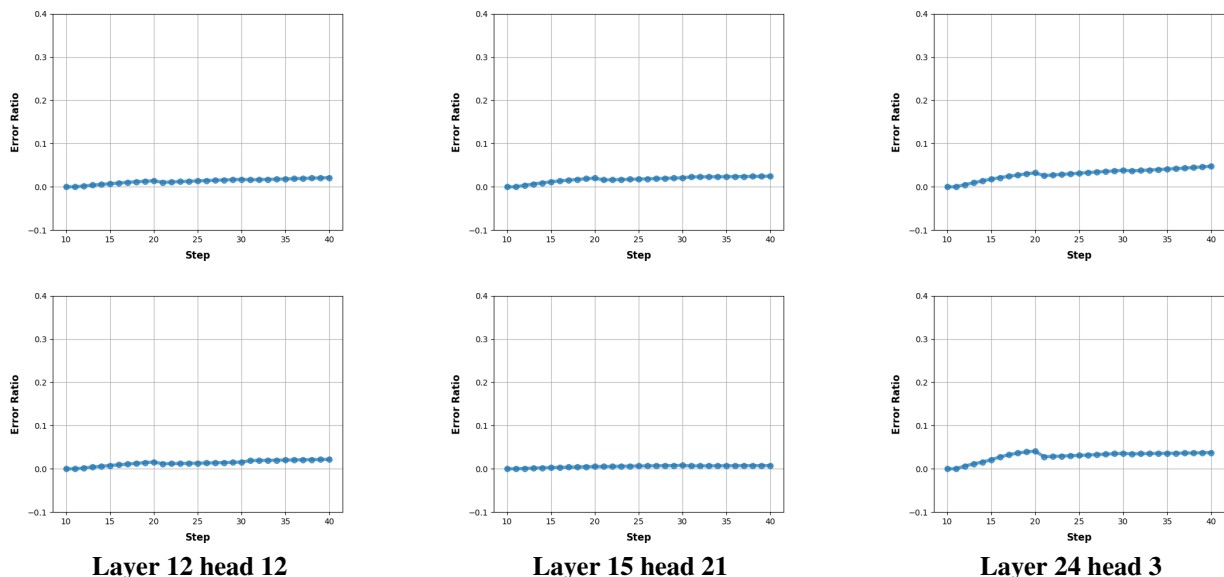

**Layer 12 head 12**     **Layer 15 head 21**     **Layer 24 head 3**

*Figure 12.* Normalized reconstruction error (NRE) results for 6 randomly sampled attention heads across varying denoising steps. Experiments are conducted under the same VBench prompt using CogVideoX-v1.5 with 49-frame 640×512 videos on NVIDIA A100 80G GPUs.

### A.1.10. EXTENDED QUALITATIVE RESULTS ON SPARSITY MASK FIDELITY

We provide extended qualitative visualizations to demonstrate MOD-DiT's capability in modeling complex attention dynamics. As illustrated in Figure 13, unlike the rigid and oversimplified structures of baseline methods, the sparsity masks generated by MOD-DiT faithfully preserve the structural characteristics of the ground-truth attention maps. This high visual fidelity confirms that MOD-DiT successfully captures the dynamic, blended attention patterns inherent to video diffusion transformers without discarding critical spatial-temporal information.

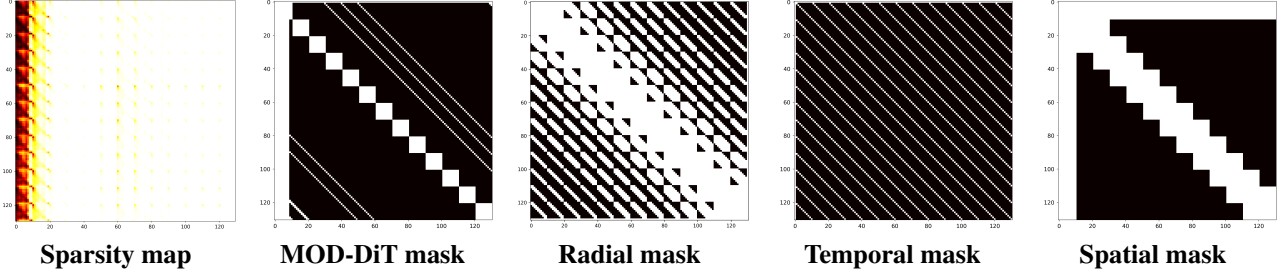

**Sparsity map**    **MOD-DiT mask**    **Radial mask**    **Temporal mask**    **Spatial mask**

*Figure 13.* Visual comparison of various sparse attention methods generate corresponding masks for specific attention heads during the inference of CogVideo(Yang et al., 2024).

### A.1.11. ABLATION STUDY ON PIECEWISE LINEARITY OF VERTICAL AND PARALLEL-TO-MAIN-DIAGONAL PATTERNS

To validate the core observation in section 4.3 that the intensities of vertical ($d_k^{(t)}$) and parallel-to-main-diagonal ($c_k^{(t)}$) patterns exhibit stable piecewise linearity in the mid-to-late denoising stage ($t > m$, $m = 12$), we conduct a systematic statistical analysis with large-scale data points.

**Experimental Setup**

- **Model**: CogVideoX-v1.5
- **Data**: 5 diverse prompts randomly selected from VBench (Huang et al., 2023), covering different motion types and

scenes, with 89-frame videos (640×512 resolution) generated for each prompt.
- **Attention Heads/Layers**: 10 layers randomly sampled, 6 attention heads per layer, resulting in $5 \times 10 \times 6 = 300$ independent data points.
- **Data Extraction**: For each prompt, layer, and head, we extract the intensity sequences of vertical/parallel-to-main-diagonal patterns in the mid-to-late denoising stage ($t \in [13, 50]$), and compute the normalized residual error (NRE) of piecewise linear fitting.

**Evaluation Metric**    We define the *Normalized Residual Error (NRE)* to quantify the linear fitting accuracy, which measures the deviation between the true intensity and the fitted value:

$$\text{NRE} = \frac{\sqrt{\frac{1}{N_t} \sum_{t \in \text{segment}} (x_k^{(t)} - \hat{x}_k^{(t)})^2}}{\max(x_k^{(t)}) - \min(x_k^{(t)})}$$

where $x_k^{(t)}$ denotes the true intensity obtained by equation (4)(vertical $d_k^{(t)}$ or parallel-to-main-diagonal $c_k^{(t)}$), $\hat{x}_k^{(t)}$ is the value obtained by linear prediction, and $N_t$ is the number of denoising steps in the segment. A smaller NRE indicates stronger linearity; we define NRE < 0.1 as "strong linearity".

**Results and Analysis**    As observed in Figure 14, the NRE distribution of 300 data points almost all fall within the interval [0.00, 0.10], which confirm that the piecewise linearity of vertical and parallel-to-main-diagonal patterns is not an isolated case but a *universal property* across multiple prompts, layers, and heads in the CogVideo model.

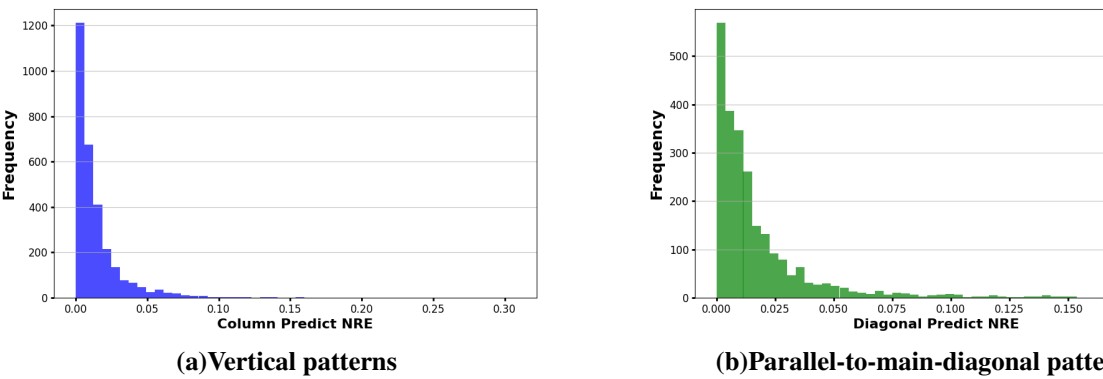

(a)Vertical patterns          (b)Parallel-to-main-diagonal patterns

*Figure 14.* Histograms of NRE for vertical and parallel-to-main-diagonal patterns, which are based on 300 data points (5 prompts × 10 layers × 6 heads).

## A.2. Additional Quantitative Results on CogVideoX-v1.5

*Table 5.* Quantitative comparison on CogVideoX-v1.5 (Yang et al., 2024) using VBench prompts. To ensure strict fairness, all sparse baselines utilize the same inference API. Model configuration: CogVideoX-v1.5 (2B parameters, 89 frames, $640 \times 512$ resolution, A800 GPU). Our method consistently achieves the best quality-efficiency trade-off, delivering maximum speedups and top scores while maintaining the **highest sparsity**.

| Model | Method | Sparsity | Quality | | | | | Efficiency | |
|---|---|---|---|---|---|---|---|---|---|
| | | | PSNR↑ | SSIM↑ | LPIPS↓ | SubConsist↑ | ImageQual↑ | Latency(s)↓ | Speedup↑ |
| CogVideoX-v1.5 | Full | 0.00% | - | - | - | 0.9230 | 0.6255 | 987s | 1× |
| | MInference | 64.9% | 15.01 | 0.601 | 0.334 | 0.8679 | 0.5580 | 696s | 1.42× |
| | Radial | 70.7% | 22.89 | 0.866 | 0.172 | 0.9214 | 0.6167 | 611s | 1.62× |
| | SVG | 75.0% | 21.15 | 0.818 | 0.183 | 0.9158 | 0.5948 | 596s | 1.65× |
| | Sparge | 67.3% | 20.34 | 0.773 | 0.255 | 0.9043 | 0.5966 | 661s | 1.49× |
| | **Ours** | **80.1%** | 25.77 | 0.868 | 0.133 | 0.9266 | 0.6239 | **542s** | **1.82×** |

### A.3. Results of MOD-DiT and SVG-2 in VBench

To further verify the superiority of our method against the latest iteration of the SVG baseline, we conduct additional comparisons with SVG-2 under strictly unified settings. Experiments are performed on the Hunyuan model with NVIDIA A100 80GB GPU, 117 frames, and 768×1280 resolution. SVG-2 is evaluated using the recommended parameters from its original paper to ensure a fair and apples-to-apples comparison.

*Table 6.* VBench results on the Hunyuan model (A100 80GB GPU, 117 frames, 768×1280 resolution), SVG-2 adopts the recommended parameters from the original paper

| Method | S.C. | B.C. | M.S. | T.F. | I.Q. | A.Q. | D.D. | latency |
|--------|------|------|------|------|------|------|------|---------|
| SVG-2 | 0.9325 | 0.9369 | 0.9636 | 0.9518 | 0.6431 | 0.5901 | 0.9076 | 1056s |
| **Ours** | 0.9398 | 0.9320 | 0.9687 | 0.9527 | 0.6587 | 0.5899 | 0.9104 | 1044s |

As shown in table 6, MOD-DiT outperforms SVG-2 on most key VBench dimensions, including subject consistency, motion smoothness, and imaging quality, while achieving slightly lower latency (1044s vs. 1056s). This confirms that our method maintains superior video generation quality even when compared to the latest iteration of the SVG baseline.

### A.4. Compatibility with Step-Distilled Diffusion Models

To verify the generalizability of MOD-DiT beyond standard full-step models, we conduct comprehensive experiments to evaluate its compatibility with **step-distilled diffusion models**, using the 8-step distilled FastWan model (a variant of Wan2.1) as the testbed. Key findings are supported by quantitative results (Table 7) and qualitative visualizations.

*Table 7.* Results of Applying MOD-DiT to the Distilled Model FastWan (8-step)

| Method | S.C. | B.C. | M.S. | T.F. | I.Q. | A.Q. | D.D. | latency |
|--------|------|------|------|------|------|------|------|---------|
| FastWan | 0.9317 | 0.9489 | 0.9766 | 0.9658 | 0.6531 | 0.5910 | 0.9201 | 327s |
| **FastWan+MOD-DiT** | 0.9287 | 0.9470 | 0.9692 | 0.9543 | 0.6502 | 0.5904 | 0.9176 | 210s |

**(D1) Attention pattern validity in distilled models**   We visualize the attention maps of FastWan during denoising inference. As shown in Figure 15, the distilled DiT model still exhibits our core three attention patterns (column, parallel-diagonal, and block patterns) throughout the inference process, proving that our fundamental assumption remains valid for step-distilled models.

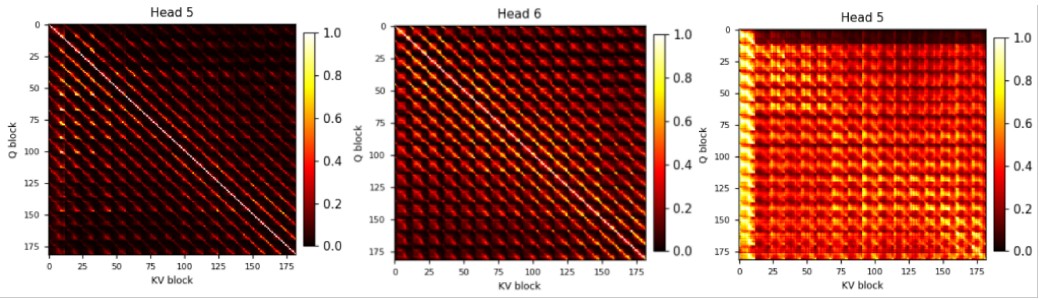

*Figure 15.* Visualization of attention patterns on the FastWan distilled model

**(D2) Pattern evolution and low approximation error**   We analyze the evolution of pattern intensities across denoising steps (Figure 16) and compute the normalized approximation error (NAE) (Figure 17). Results show extremely low NAE values across all denoising steps, verifying the high accuracy of our three-pattern linear approximation on distilled models.

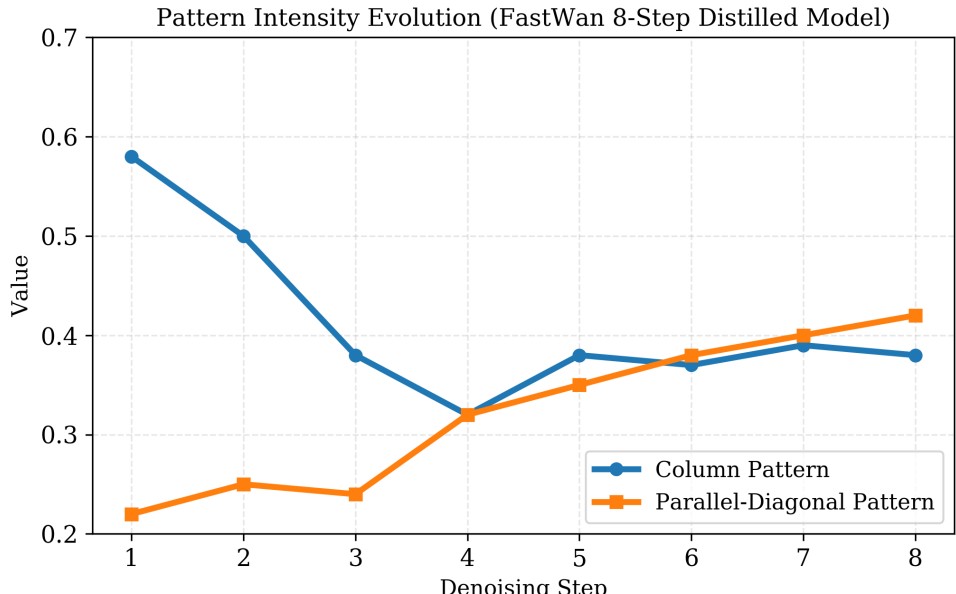

*Figure 16.* Intensity evolution of two core attention patterns on FastWan

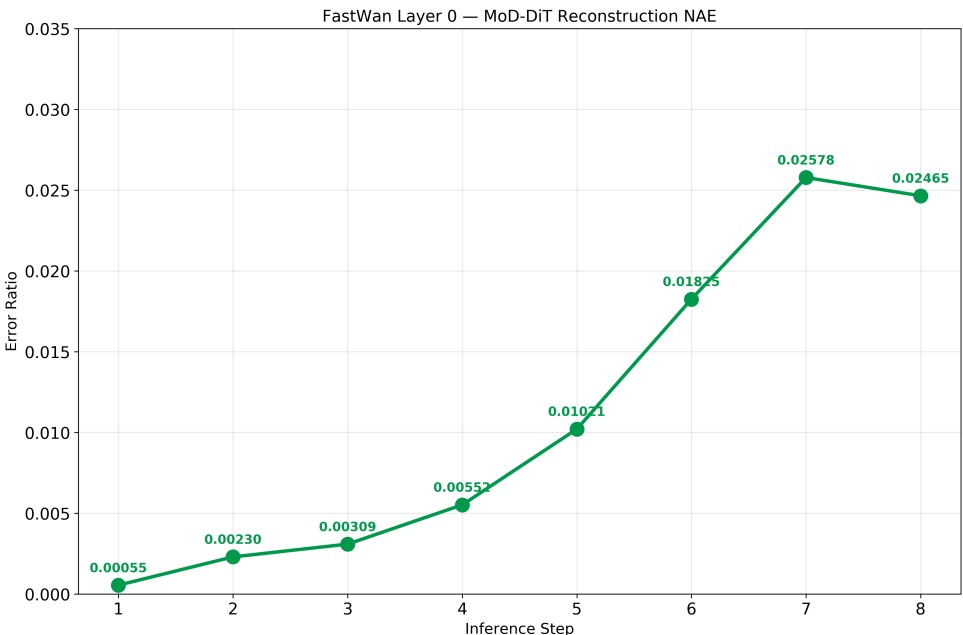

*Figure 17.* Normalized approximation error (NAE) across denoising steps on FastWan

**(D3) End-to-end acceleration results**   As demonstrated in Table 7, MOD-DiT achieves an additional $1.56\times$ speedup on the distilled FastWan model (reducing latency from 327s to 210s), with only negligible performance drop on all VBench quality metrics. This confirms that MOD-DiT can further accelerate optimized step-distilled models without sacrificing generation quality.

In conclusion, MOD-DiT maintains strong compatibility with step-distilled diffusion models, extending its acceleration capability to both full-step and distilled video generation pipelines.

## A.5. Video Generation Results of MOD-DiT

To visually validate MOD-DiT's ability to balance generation quality and inference efficiency across diverse inputs and models, Figures 18, 19, and 20 show qualitative comparisons of its visualization results against other sparse attention methods (e.g., SVG(Xi et al., 2025), Radial Attention(Li et al., 2025)) on Hunyuan Video(Kong et al., 2024) and Wan 2.1(Wang et al., 2025). On HunyuanVideo (Figure 1), MOD-DiT achieves a consistent $2.05\times$ speedup with outputs nearly identical to full attention—no visible loss in details, spatiotemporal coherence, or color fidelity. On Wan 2.1 (Figures 19 and 20, processing 81-frame 512×832 videos), it maintains full-attention-level subject consistency and scene realism while delivering a $1.75\times$ speedup. These comparisons confirm MOD-DiT retains full-attention-quality generation with substantial acceleration across different models and inputs, showcasing strong adaptability.

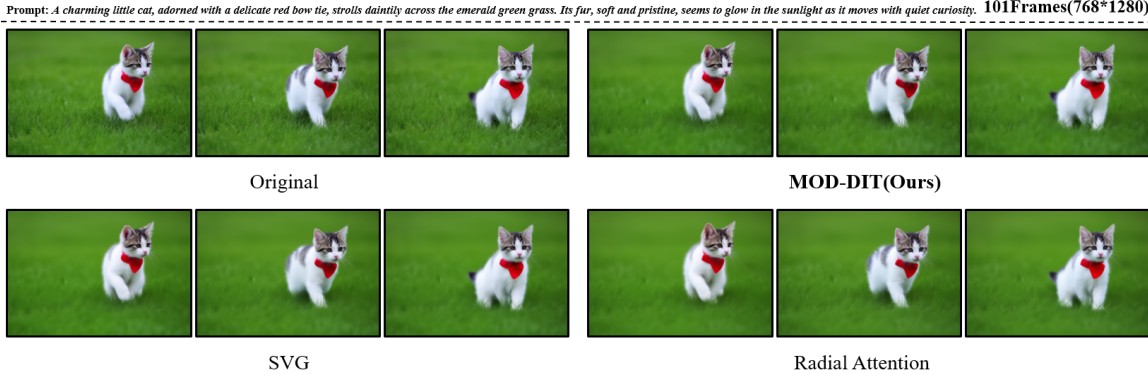

*Figure 18.* Comparison of the visualization effects of different sparse attention methods on HunyuanVideo(Kong et al., 2024). Our method MOD-DiT consistently achieves $2.05\times$ speedup, and keep almost the same as original videos.

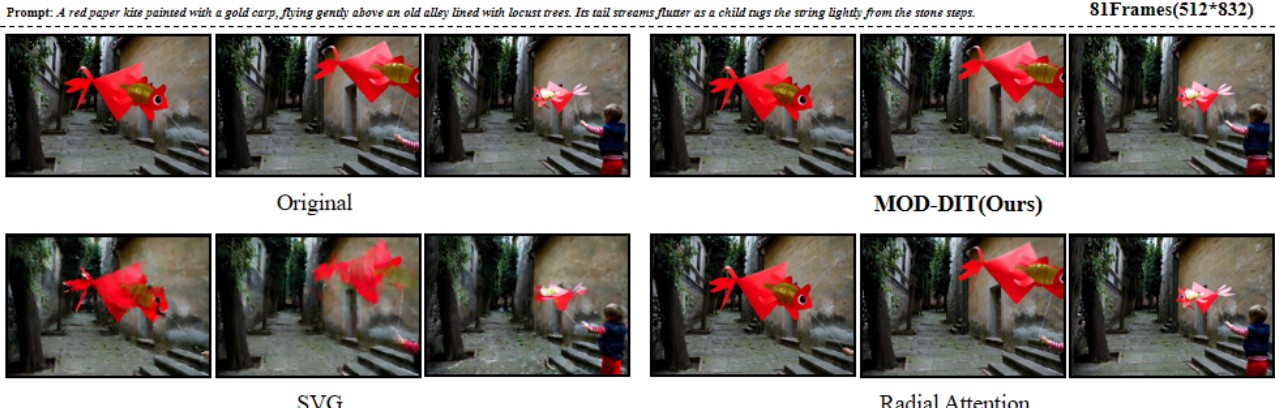

*Figure 19.* Comparison of the visualization effects of different sparse attention methods on Wan 2.1(Wang et al., 2025). Our method MOD-DiT consistently achieves $1.75\times$ speedup, and keep almost the same as original videos.

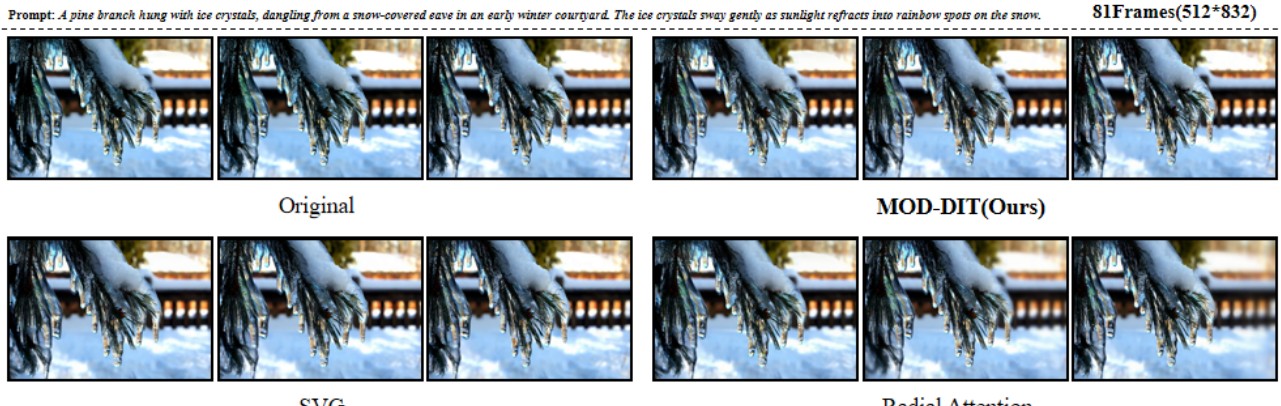

*Figure 20.* Comparison of the visualization effects of different sparse attention methods on Wan 2.1(Wang et al., 2025). Our method MOD-DiT consistently achieves $1.75\times$ speedup, and keep almost the same as original videos.

---

**Algorithm 1** MOD-DiT: Dynamic Sparse Attention for Efficient Video Diffusion

---

**Require:** Input features $I \in \mathbb{R}^{B_b \times N \times D}$, Number of heads $H$, Total denoising steps $T$, Warm-up steps $m$, Prediction interval $\Delta t$, Block size $B$, Sparsity threshold $\eta$, Block-diagonal threshold $\tau_e$, Top-K value $K$

**Ensure:** Generated video features after efficient diffusion inference

1: Linear projection of input $I$ to get query $Q$, key $K$, value $V$: $Q, K, V \in \mathbb{R}^{B_b \times H \times N \times d}$ (where $d = D/H$)
2: **Warm-up Phase:**
3: **for** $t = 1$ **to** $m$ **do**
4:     Compute full attention map $A^{(t)} = \text{softmax}\left(\frac{QK^T}{\sqrt{d}}\right)$ using FlashAttention-2 (Dao, 2023)
5: **end for**
6: $\hat{A}^{(m-1)} = A^{(m-1)}$, $\hat{A}^{(m)} = A^{(m)}$
7: Convert $\hat{A}^{(m-1)}$ to sparsity map $\hat{S}^{(m-1)} \in \mathbb{R}^{(N/B) \times (N/B)}$ via Eq.(2): partition $\hat{A}^{(m-1)}$ into non-overlapping $B \times B$ blocks, calculate block sparsity with $\eta$
8: Convert $\hat{A}^{(m)}$ to sparsity map $\hat{S}^{(m)} \in \mathbb{R}^{(N/B) \times (N/B)}$ via Eq.(2), calculate block sparsity with $\eta$
9: Compute $X^{(m-1)} = [c_k^{(m-1)}, d_k^{(m-1)}, e_k^{(m-1)}]$ via Eq.(4) and least-squares kernel (Sec.B)
10: Compute $X^{(m)} = [c_k^{(m)}, d_k^{(m)}, e_k^{(m)}]$ via Eq.(4) and least-squares kernel (Sec.B)
11: Predict initial intensities $\{\hat{c}_k^{(t')}, \hat{d}_k^{(t')}\}$ for $t' \in (m, m + \Delta t]$ using linear interpolation (Sec.5.2)
12: **Main Denoising Phase:**
13: **for** $t = m + 1$ **to** $T$ **do**
14:     **if** $t \bmod \Delta t == 0$ **then**
15:         Reconstruct $\hat{A}^{(t)}$ by fusing masked $A_{\text{masked}}^{(t)}$ (from current sparse attention) and historical $\hat{A}^{(t-\Delta t)}$ (Sec.5.1)
16:         Convert $\hat{A}^{(t)}$ to sparsity map $\hat{S}^{(t)} \in \mathbb{R}^{(N/B) \times (N/B)}$ via Eq.(2)
17:         Compute intensities $X^{(t)} = [c_k^{(t)}, d_k^{(t)}, e_k^{(t)}]$ via Eq.(4) and least-squares kernel (Sec.B)
18:         Predict intensities $\{\hat{c}_k^{(t')}, \hat{d}_k^{(t')}\}$ for $t' \in (t, t + \Delta t]$ using linear interpolation (Sec.5.2)
19:     **end if**
20:     For vertical/parallel-diagonal patterns: Select Top-K informative patterns from $\{\hat{c}_k^{(t)}, \hat{d}_k^{(t)}\}$ (Sec.5.3), map to $B \times B$ block positions
21:     For block-diagonal patterns: Preserve if $\min(\hat{e}^{(m-1)}, \hat{e}^{(m)}) > \tau_e$ (Sec.5.3), where block-diagonal regions are defined as $B \times B$ blocks (consistent with block size $B$)
22:     Construct block-wise mask $M^{(t)} \in \mathbb{R}^{(N/B) \times (N/B)}$: mark valid/invalid $B \times B$ blocks by combining Top-K patterns and preserved block-diagonal patterns
23:     Upsample $M^{(t)}$ to token-level mask $\bar{M}^{(t)} \in \mathbb{R}^{N \times N}$: extend each block's mask state to all $B \times B$ tokens in the block
24:     Concatenate masks across all attention heads: $\bar{M}^{(t)} = \text{cat}(\bar{M}_1^{(t)}, \bar{M}_2^{(t)}, \ldots, \bar{M}_H^{(t)})$
25:     Compute block-wise sparse attention via Eq.(1) (SageAttention (Zhang et al., 2025b)):
26:         1. Split $Q, K$ into $B \times B$ token blocks (consistent with block size $B$)
27:         2. Apply $\bar{M}^{(t)}$ to filter invalid blocks, skip redundant token interactions
28:         3. Online softmax on valid blocks: $O = \text{softmax}\left(\frac{QK^T}{\sqrt{d}} + \bar{M}^{(t)}\right) V$
29:     Update features with sparse attention output $O$
30: **end for**
31: **Return** Final video features after $T$ denoising steps

---

# B. Comprehensive Design of the Efficient Least Squares Kernel for Dynamic Attention Pattern Approximation

This section provides a detailed exposition of the optimized least squares kernel designed for efficient approximation of dynamic attention patterns in MOD-DiT. The kernel addresses the fundamental computational challenges through sophisticated mathematical formulation, advanced parallel computation strategies, and robust numerical optimization techniques, enabling real-time inference while maintaining high accuracy.

## B.1. Key Notations

To ensure clarity, we first supplement the key notations used in this section (consistent with the main text):

- $S^{(t)} \in \mathbb{R}^{n \times n}$: Attention sparsity map at denoising step $t$, derived from partitioning the full attention map into $B \times B$ non-overlapping blocks.

- $n$: Number of blocks per dimension in the sparsity map, where $n = N/B$ ( $N$ is total number of tokens, $B$ is block size).

- $M \in \mathbb{R}^{n^2 \times (3n-1+|\mathcal{A}|)}$: Design matrix composed of basis matrices for three core attention patterns.

- $X^{(t)} \in \mathbb{R}^{(3n-1+|\mathcal{A}|) \times 1}$: Coefficient vector to be optimized, including intensities of parallel-to-main-diagonal, vertical, and block-diagonal patterns.

- $\{C_k\}_{k=1}^{2n-1}$: Basis matrices for parallel-to-main-diagonal patterns, with $\delta_k = k - (n-1)$ denoting the diagonal offset.

- $\{D_k\}_{k=1}^{n}$: Basis matrices for vertical patterns.

- $\{E_k\}_{k \in \mathcal{A}}$: Basis matrices for block-diagonal patterns, where $\mathcal{A}$ is the set of block-diagonal indices.

- $|\mathcal{A}|$: Number of block-diagonal basis matrices, determined by the block partition strategy.

- $H$: Number of attention heads in the transformer architecture.

- $b$: Number of tokens per frame in video generation.

- $m$: Number of frames, where $m = n/b$.

## B.2. Design Insight and Motivation

The core challenge addressed by this kernel is the efficient approximation of dynamic attention patterns in video diffusion transformers. As revealed in the main text, attention sparsity maps $S^{(t)}$ exhibit a dynamic mixture of three structured patterns (parallel-to-main-diagonal, vertical, block-diagonal) that evolve across denoising steps. Directly solving the least squares problem for pattern approximation would incur prohibitive computational costs: - Naive construction of the design matrix $M$ requires $O(n^2 \times (3n - 1 + |\mathcal{A}|))$ storage, which is infeasible when $n$ reaches thousands of tokens. Explicit matrix operations for least squares solution would result in $O(n^5)$ time complexity, making real-time inference impossible.

To address these issues, MOD-DiT's kernel is designed with three core insights.

**Structural Sparsity Exploitation.** The three core attention patterns have inherent sparse and regular structures, enabling analytical computation of matrix products (e.g., $M^T M$) without explicit matrix construction.

**Computational Complexity Reduction.** By decomposing high-complexity matrix operations into low-dimensional structured computations, the kernel reduces time complexity from naive $O(Hn^5)$ to $O(Hn^3)$ and storage complexity from $O(n^3)$ to $O(n^2)$.

**Hardware-Aware Optimization.** Leveraging GPU multi-level parallelism (inter-head, inter-feature, element-wise) optimizes memory access and computation efficiency, ensuring the kernel overhead accounts for only 1-2% of total attention computation time.

This design enables MOD-DiT to achieve $100\times$ speedup in solving the least squares problem compared to native solvers (e.g., PyTorch's `torch.lstsq`), while maintaining high approximation accuracy for dynamic attention patterns—critical for balancing inference efficiency and video generation quality.

## B.3. Theoretical Foundation and Problem Formulation

The core mathematical problem involves approximating the attention sparsity map $S^{(t)} \in \mathbb{R}^{n \times n}$ at denoising step $t$ through a linear combination of structured basis matrices. The objective function is formulated as the minimization of the Frobenius norm:

$$\min_{X^{(t)}} \left\| vec(S^{(t)}) - MX^{(t)} \right\|_F^2$$

where $vec(\cdot)$ denotes matrix vectorization, $M$ is the design matrix integrating the three core pattern families, and $X^{(t)}$ is the coefficient vector quantifying the intensity of each pattern at step $t$. The Frobenius norm ensures optimal fitting in the matrix space while maintaining numerical stability.

The design matrix $M$ incorporates three distinct pattern types that capture the essential characteristics of attention mechanisms in video generation:

- **Parallel-to-main-diagonal patterns** $\{C_k\}_{k=1}^{2n-1}$ model inter-frame spatial correlation, defined as:

$$C_k(i,j) = \begin{cases} 1, & \text{if } j - i = k - (n-1) \\ 0, & \text{otherwise} \end{cases}$$

where $k \in [1, \ldots, 2n-1]$ covers all possible parallel-to-main diagonals, with $\delta_k = k - (n-1)$ representing the diagonal offset.

- **Vertical patterns** $\{D_k\}_{k=1}^n$ capture global cross-token dependencies through column-wise structures:

$$D_k(i,j) = \begin{cases} 1, & \text{if } j = k \\ 0, & \text{otherwise} \end{cases}$$

- **Block-diagonal patterns** $\{E_k\}_{k \in \mathcal{A}}$ maintain intra-frame temporal coherence by grouping tokens within frames. For the $k$-th block region $B_k = [a_k, b_k] \times [a_k, b_k]$, the basis matrix is defined as:

$$E_k(i,j) = \begin{cases} 1, & \text{if } (i,j) \in B_k \\ 0, & \text{otherwise} \end{cases}$$

where $b$ denotes the number of tokens per frame, and $m = n/b$ represents the number of frames.

The complete mathematical representation of the design matrix $M$ and coefficient vector $X^{(t)}$ is:

$$M = \left[ \{vec(C_k)\}_{k=1}^{2n-1}, \{vec(D_k)\}_{k=1}^n, \{vec(E_k)\}_{k \in \mathcal{A}} \right]$$

$$X^{(t)} = \left[ c_1^{(t)}, ..., c_{2n-1}^{(t)}, d_1^{(t)}, ..., d_n^{(t)}, e_1^{(t)}, ..., e_{|\mathcal{A}|}^{(t)} \right]^T$$

where $c_k^{(t)}$, $d_k^{(t)}$, and $e_k^{(t)}$ are intensity scalars quantifying the contribution of each parallel-to-main-diagonal, vertical, and block-diagonal pattern at step $t$, respectively.

The normal equation derived from the optimization problem provides the computational foundation:

$$M^T M X^{(t)} = M^T vec(S^{(t)})$$

This formulation transforms the pattern approximation problem into solving a linear system, where the structured nature of the basis matrices enables significant computational optimizations.

## B.4. Computational Complexity Analysis and Optimization Strategies

Direct implementation of the least squares solution would require constructing the design matrix $M \in \mathbb{R}^{n^2 \times (3n-1+|\mathcal{A}|)}$, incurring $O(n^3)$ storage complexity and $O(n^5)$ time complexity for explicit matrix operations. Such computational demands are prohibitive for video generation tasks where $n$ can reach several thousand tokens.

MOD-DiT's optimization strategy leverages the inherent sparsity and structural properties of the basis matrices to achieve $O(n^2)$ storage complexity and $O(n^3)$ time complexity through analytical block decomposition. The $M^T M$ matrix exhibits a block structure that enables efficient computation:

$$M^T M = \begin{bmatrix} C^T C & C^T D & C^T E \\ D^T C & D^T D & D^T E \\ E^T C & E^T D & E^T E \end{bmatrix}$$

where $C = [C_1, \ldots, C_{2n-1}]$, $D = [D_1, \ldots, D_n]$, and $E = \begin{bmatrix} E_1, \ldots, E_{|\mathcal{A}|} \end{bmatrix}$. Each block can be computed analytically using specialized algorithms:

For the parallel-to-main-diagonal block $C^T C$, the elements are computed as:

$$\left[ C^T C \right]_{ij} = \langle C_i, C_j \rangle_F = \begin{cases} n - |\delta_i|, & \text{if } i = j \\ 0, & \text{otherwise} \end{cases}$$

where $\delta_i = i - (n-1)$. This results in a diagonal matrix with elements representing the lengths of corresponding parallel-to-main diagonals.

The vertical line block $D^T D$ simplifies to:

$$\left[ D^T D \right]_{ij} = \begin{cases} n, & \text{if } i = j \\ 0, & \text{otherwise} \end{cases}$$

The cross-term block $C^T D$ is computed as:

$$\left[ C^T D \right]_{ij} = \begin{cases} 1, & \text{if parallel-to-main-diagonal } i \text{ passes through column } j \\ 0, & \text{otherwise} \end{cases}$$

The block-diagonal term $E^T E$ computes as:

$$E^T E = \text{diag}\left( b^2, b^2, ..., b^2 \right) \in \mathbb{R}^{|\mathcal{A}| \times |\mathcal{A}|}$$

where each diagonal element equals $b^2$ (the number of elements in each block-diagonal region).

The right-hand side vector $M^T vec(S^{(t)})$ is computed through partitioned inner products:

$$M^T vec(S^{(t)}) = \begin{bmatrix} C^T vec(S^{(t)}) \\ D^T vec(S^{(t)}) \\ E^T vec(S^{(t)}) \end{bmatrix}$$

where each component is calculated efficiently:

$$\left[ C^T vec(S^{(t)}) \right]_k = \sum_{(i,j) \in \mathcal{D}_k} S^{(t)}(i,j),$$

$$\left[ D^T vec(S^{(t)}) \right]_j = \sum_{i=0}^{n-1} S^{(t)}(i,j),$$

$$\left[ E^T vec(S^{(t)}) \right]_k = \sum_{(i,j) \in B_k} S^{(t)}(i,j)$$

with $\mathcal{D}_k$ denoting the set of indices for the $k$-th parallel-to-main-diagonal pattern.

### B.5. Multi-level Parallelization Architecture and Kernel Implementation

The kernel implements a sophisticated three-level parallelization strategy optimized for modern GPU architectures, ensuring maximal hardware utilization and computational efficiency.

**First Level: Inter-head Parallelism** leverages the independence between attention heads by distributing computations across multiple GPU streaming multiprocessors. Each attention head $h \in [1, H]$ is processed concurrently, with head indices mapped to the y-dimension of the execution grid. This level exploits the natural parallelism in transformer architectures where attention heads operate independently.

**Second Level: Inter-feature Parallelism** assigns different geometric features (parallel-to-main-diagonal, vertical, block-diagonal) to separate thread blocks for simultaneous execution. This approach maximizes occupancy by utilizing the GPU's ability to manage multiple concurrent thread blocks, with feature assignments optimized to balance computational load.

**Third Level: Element-wise Parallelism** employs fine-grained parallel reduction models within thread blocks for efficient summation operations. Using a tree-based reduction pattern with 256-thread blocks, this level ensures optimal memory access patterns and computational density.

The kernel design incorporates specialized functions for different computational patterns.

INNER PRODUCT CALCULATION KERNEL

Implements the Frobenius inner product for matrices $A, B \in \mathbb{R}^{n \times n}$:

$$\langle A, B \rangle_F = \sum_{i=1}^{n} \sum_{j=1}^{n} A(i, j) \cdot B(i, j) = \mathrm{tr}(A^T B)$$

The kernel flattens matrices into vectors $v \in \mathbb{R}^{n^2}$ and employs a hierarchical reduction strategy:

$$s_t^{(0)} = \sum_{k=t}^{n^2} v_k^2, \quad k \equiv t \pmod{T}$$

where $T = 256$ is the thread block size. The reduction follows a logarithmic pattern:

$$s_t^{(\ell+1)} = s_t^{(\ell)} + s_{t+2^\ell}^{(\ell)}, \quad \ell = 0, 1, \ldots, \log_2(T) - 1$$

with results combined atomically across thread blocks.

PARALLEL-TO-MAIN-DIAGONAL SUMMATION KERNEL

Processes parallel-to-main-diagonal patterns with optimal memory access patterns. For pattern $k$ with offset $\delta_k = k - (n-1)$:

$$\mathcal{D}_k = \begin{cases} \{(i, i + \delta_k) : i \in [0, n - 1 - \delta_k]\}, & \delta_k \geq 0 \\ \{(i - \delta_k, i) : i \in [0, n - 1 + \delta_k]\}, & \delta_k < 0 \end{cases}$$

The kernel employs efficient index mapping and shared memory reduction to compute:

$$\langle S_h^{(t)}, C_k \rangle = \sum_{(i,j) \in \mathcal{D}_k} S_h^{(t)}(i, j)$$

VERTICAL LINE SUMMATION KERNEL

Computes column sums through parallel accumulation:

$$\langle D_k, S_h^{(t)} \rangle = \sum_{i=0}^{n-1} S_h^{(t)}(i, k) = \mathbf{1}^T S_h^{(t)} e_k$$

where $e_k$ is the standard basis vector.

BLOCK SUMMATION KERNEL

Handles block-diagonal patterns through partitioned computation:

$$\langle E_k, S_h^{(t)} \rangle = \sum_{(i,j) \in B_k} S_h^{(t)}(i,j)$$

with each block processed concurrently using optimized memory access patterns.

### B.6. Numerical Stability and Linear System Solution Techniques

To ensure numerical robustness in solving the linear system $M^T M X^{(t)} = M^T vec(S^{(t)})$, MOD-DiT incorporates several advanced techniques:

REGULARIZATION SCHEME

A Tikhonov regularization approach ensures well-posedness:

$$M^T M \leftarrow M^T M + \lambda I$$

where $\lambda = 10^{-8}$ provides stability without significantly affecting solution quality. The regularization parameter is chosen through empirical analysis to balance numerical stability and approximation accuracy.

ADAPTIVE SOLUTION STRATEGIES

The system employs multiple solution methods adaptively based on matrix properties:

**Cholesky Decomposition.** Primary method for positive definite systems:

$$M^T M = LL^T, \quad X^{(t)} = L^{-T}\left(L^{-1}(M^T vec(S^{(t)}))\right)$$

This approach offers optimal numerical stability with $O(n^3)$ complexity for the factorization.

**LU Decomposition with Partial Pivoting.** Fallback method for numerically challenging systems:

$$P(M^T M) = LU, \quad X^{(t)} = U^{-1}\left(L^{-1}P(M^T vec(S^{(t)}))\right)$$

where $P$ is the permutation matrix for numerical stability.

**Moore-Penrose Pseudoinverse.** Handles rank-deficient cases:

$$X^{(t)} = (M^T M)^{\dagger} M^T vec(S^{(t)})$$

computed via singular value decomposition with threshold-based singular value retention.

### B.7. Complexity Analysis and Performance Characterization

The overall computational complexity of MOD-DiT's kernel is analyzed through meticulous examination of each component:

- **Matrix Product Computation**: $M^T M$ calculation requires $O(Hn^2)$ operations leveraging structural sparsity.

- **Right-hand Side Computation**: $M^T vec(S^{(t)})$ evaluation consumes $O(Hn^2)$ operations through optimized summations.

- **Linear System Solution**: The factorization and solve steps require $O(Hn^3)$ operations.

The total complexity of $O(Hn^3)$ represents a significant improvement over the naive $O(Hn^5)$ approach. MOD-DiT's optimized implementation achieves demonstrated speedups of 100× compared to standard solvers like `torch.lstsq`.

## C. Computation Analysis

### C.1. Additional computation experiments results

We conducted a comparative experiment to evaluate the computational costs of various MOD-DiT operations against a single Full Attention operation. As shown in table8, the additional computational overhead introduced by MOD-DiT is negligible compared to Full Attention. Moreover, when masking 50% of the blocks, MOD-DiT's computational cost is only 74% of that of Full Attention.

*Table 8.* Comparison of computational costs between MOD-DiT additional operations and one Full Attention operation using the hunyuan model with size $512 \times 832 \times 81$.

| Operation Description | FLOPS (TFLOPS) |
|---|---|
| Full Attention | 7.6127 |
| Convert to Sparsity | 0.0297 |
| Solve kernel | 0.0002 |
| MOD-DiT: topk200 (50% mask) | 5.7096 |

### C.2. Theoretical Analysis

We conduct a detailed theoretical analysis of the additional computational overhead introduced by MOD-DiT's core components, focusing on a single attention head to clarify complexity contributions. All overheads are compared against the computational cost of full attention (denoted as $O(N^2D)$, where $N$ is the total number of tokens and $D$ is the feature dimension per token) to verify their negligibility.

#### C.2.1. OVERHEAD OF ATTENTION-TO-SPARSITY MAP TRANSFORMATION

The transformation from the attention map $A$ to the attention sparsity map $S$ (Eq. (2) in the main text) involves partitioning $A$ into non-overlapping $B \times B$ blocks and calculating the sparsity scalar for each block. For a single attention head, this operation only requires a single traversal of all elements in the attention map. Each element is checked against the sparsity threshold $\eta$ once, and the sum of valid elements per block is computed. The total computational complexity of this step is $O(N^2)$, as it scales linearly with the number of elements in the attention map.

#### C.2.2. OVERHEAD OF SPARSITY MAP RECONSTRUCTION

Sparsity map reconstruction (Sec.5.1) fuses the masked attention map at the current step with the reconstructed complete attention map from the previous step. For each token pair $(p, q)$, the reconstruction operation simply selects the valid attention value based on the mask state (either retaining the current masked value or reusing the historical reconstructed value). This process requires traversing all elements of the attention map exactly once, resulting in a computational complexity of $O(N^2)$ for a single attention head.

#### C.2.3. OVERHEAD OF LINEAR PREDICTION AND TOP-K ROUTER

**Linear Prediction.** The linear prediction module (Sec. 5.1) models the intensity scalars of parallel-diagonal and vertical patterns as linear functions of denoising steps. The total number of prediction steps is $\frac{T}{\Delta t}$, where $T$ is the total number of denoising steps and $\Delta t$ is the interval between consecutive sparsity map updates. Each prediction step leverages precomputed intensity scalars from historical steps and performs a constant-time linear interpolation, leading to an overall complexity of $O\left(\frac{T}{\Delta t}\right)$.

**Top-K Router.** The Top-K selection (Sec. 5.3) sorts the merged intensity scalars of parallel-diagonal and vertical patterns (totaling $O(n)$ elements, where $n = N/B$ is the number of blocks per dimension) and retains the top $K$ informative patterns. The time complexity of sorting $O(n)$ elements for Top-K selection is $O(n \log K)$, which is efficient given the typical range of $K$ (much smaller than $n$).

### C.3. Aggregate Overhead and Negligibility

Summing the additional computational overheads for a single attention head, the dominant term is $O(N^2)$. The remaining terms $O\left(\frac{T}{\Delta t}\right)$ (linear prediction) and $O(n \log K)$ (Top-K router) are sub-quadratic in $N$ (e.g., $n = N/B$ reduces scale, $\frac{T}{\Delta t}$ is a fixed constant for inference) and thus negligible compared to $O(N^2)$.

Compared to the computational cost of full attention ($O(N^2 D)$), the additional overhead is theoretically negligible. In practical diffusion transformer (DiT) architectures (e.g., DiT-Base/ Large, where $D = 768/1024$; video-specific variants like CogVideoX-v1.5 often use $D \geq 768$), $D$ is typically a large value (hundreds to thousands). This makes $N^2 D$ vastly larger than $N^2$, leading the additional overhead to account for only 1–2% of full attention's computation cost—consistent with experimental measurements.

