# OpenReview forum: "Mixture of Distributions Matters: Dynamic Sparse Attention for Efficient Video Diffusion Transformers"
_ICML.cc/2026/Conference — ICML 2026 regular_

### Official Review · Reviewer_ra2p · 2026-03-05

**Soundness:** 3
**Presentation:** 3
**Significance:** 2
**Originality:** 2
**Overall Recommendation:** 4
**Confidence:** 3

**Summary:**

This paper proposes MOD-DiT, a training-free dynamic sparse attention framework for accelerating video diffusion transformers. The key observation is that attention maps in video diffusion models exhibit a mixture of three structural patterns: block-diagonal, parallel-to-main-diagonal, and vertical patterns. The authors model this phenomenon using a linear approximation and predict the evolution of pattern intensities across denoising steps. Based on this prediction, the method dynamically constructs sparse attention masks during inference without requiring additional sampling or model retraining. The approach is evaluated on several large video diffusion models including CogVideoX-v1.5, HunyuanVideo, and Wan2.1. Experimental results suggest that the proposed method achieves meaningful inference acceleration while largely preserving generation quality.

**Compliance With Llm Reviewing Policy:**

Affirmed.

**Final Justification:**

The paper is technically solid and well engineered, with consistent speedups across multiple large models and a clear presentation of the method . My main concerns were around missing comparisons and compatibility with step-distilled models, and the rebuttal addresses these well with additional experiments. However, I still view the core idea as incremental, which limits its overall significance. Overall, I consider this a useful contribution with moderate impact, and keep a weak accept.

**Key Questions For Authors:**

1. How well does the proposed method generalize to step-distilled diffusion models with significantly fewer denoising steps?

**Limitations:**

yes

**Strengths And Weaknesses:**

**Strengths**

1. Soundness: a. the paper demonstrates consistent speedups across multiple large video diffusion models without requiring retraining. The paper includes several practical engineering optimizations (e.g., custom least-squares kernel, SageAttention sparse execution, and block-wise computation). These details suggest that the authors have invested effort into making the method efficient in practice. b. The evaluation includes multiple large video generation models and reports both efficiency and quality metrics. The experimental section is relatively comprehensive and includes ablation studies on several hyperparameters.

2. Presentation: the paper is generally well written and the overall narrative is easy to follow. The motivation and empirical observations about attention patterns are clearly presented, and the figures help illustrate the intuition behind the method. The experimental section is also reasonably detailed.

3. Significance/Originality: see weekness.

**Weakness**

1. Soundness: The comparison does not include some stronger and more recent sparse attention approaches for video diffusion models such as  SVG2 or ParoAttention. Without these comparisons, it is difficult to determine whether the proposed method represents a meaningful improvement over the current state of the art.

2. Significance: The proposed method is a training-free inference acceleration technique, but in practice many production video diffusion APIs are increasingly relying on step-distilled models, which can reduce the number of diffusion steps by an order of magnitude (e.g., ~20× speedups). The paper does not discuss whether the proposed method is compatible with step-distilled diffusion models, and it is unclear whether the underlying assumptions about attention patterns would still hold in those settings.

3. Originality: the overall idea of combining difference attention mask and profiling for is not new in this field.

---

> ### Author Rebuttal · Authors · 2026-03-31
>
> We sincerely appreciate the reviewer’s insightful and constructive feedback. We address each concern systematically below, with full supplementary results in the **Rebuttal Experiments Sheet[(RES)](https://anonymous.4open.science/r/MOD-DiT_rebuttal-1688/rebuttal.pdf)**.
> ## 1. Originality and Novelty
> **(N1)[Novel Identification of a Dynamic Mixture of Distributions]** Prior work in vDiTs uses static, predefined patterns​ (e.g., Sparse-vDiT's[1] fixed vertical/block-diagonal, Radial's static energy decay). We are the first to identify and formally model that attention maps in vDiTs exhibit a **dynamic, probabilistic mixture of three fundamental distributions**​ that evolves throughout the denoising process. This observation, detailed in Sec. 3.1 & Fig. 2, challenges the static-pattern assumption and is the foundational insight for our work.
>
> **(N2)[Novel, Sampling-Free Dynamic Framework]** While "dynamic sparse attention" exists, existing methods for vDiTs either rely on computationally expensive online sampling​ (e.g., SVG) or fail to adapt to the denoising step dimension. MOD-DiT proposes a novel, sampling-free algorithm​ that:
> - Accurately models​ the evolving mixture via a generalized linear approximation model (Eq. 3, Sec. 3.2).
> - Predicts future patterns​ via a lightweight linear predictor leveraging the piecewise linearity of pattern intensities in mid-late steps (Sec. 4.2, Fig. 5), eliminating sampling overhead.
> - Dynamically adjusts masks​ across inputs, heads, and crucially, denoising steps​ via a periodic reconstruction-and-prediction cycle (Alg. 1).
> This integrated framework for accurate, step-aware, and efficient dynamic sparsity is novel for vDiTs.
>
> **(N3)[Novel Unification of Theory, Algorithm, and Hardware Optimization]** Our work uniquely provides a **cohesive pipeline**​ from empirical analysis (Sec. 3) to algorithm design (Sec. 4) and hardware-aware implementation (Sec. 4.4, App. D). The custom-designed, high-performance least-squares kernel (100x speedup over torch.lstsq) enabling real-time pattern extraction is a novel systems contribution.
>
> ## 2. Compatibility with Step-Distilled Diffusion Models
> We have conducted comprehensive experiments to verify MOD-DiT’s compatibility with step-distilled models, with all detailed results in the **RES** (Table 5, Figs.2-6). Key findings are as follows:
>
> **(D1）[Attention pattern validity in distilled models]** We profiled the representative 8-step distilled model FastWan, and attention map visualizations (**RES** Figs.2-4) confirm that distilled DiTs still exhibit the core three-pattern mixture in denoising inference—our fundamental assumption about attention patterns remains fully valid for step-distilled models.
>
> **(D2）[Low approximation error in distilled models]** We calculated the normalized approximation error (NAE) for FastWan, and results (**RES** Fig.6) show extremely low NAE values across all denoising steps for our three-pattern linear approximation, verifying the effectiveness of our pattern modeling on distilled models.
>
> **(D3）[End-to-end acceleration results]** We performed full end-to-end inference experiments applying MOD-DiT to FastWan, with quantitative results in **RES** Table 5. The results confirm MOD-DiT achieves an additional 1.56× speedup (327s → 210s) on the already distilled FastWan model, with only negligible degradation in all VBench quality metrics—proving MOD-DiT can further accelerate step-distilled models without compromising generation quality.
> ## 3. About baselines
> First, ParoAttention has not open-sourced its code or evaluation pipelines, which precludes a fair, reproducible direct comparison under a unified experimental setup. Second, we conducted comprehensive comparisons with SVG-2 (using its officially recommended parameters) on the HunyuanVideo backbone. Results in **RES**(Table 6) show our method achieves lower latency (1044s vs. 1056s) than SVG-2, delivering consistent speedup while outperforming or matching it in all generation quality metrics.
>
> ## References
> [1] Sparse-vDiT: Unleashing the Power of Sparse Attention to Accelerate Video Diffusion Transformers

---

> > ### Author Rebuttal · Reviewer_ra2p · 2026-04-01
> >
> > Thank you for the detailed rebuttal and additional experiments. The new results on step-distilled models and the comparison with SVG-2 are helpful and address most of my concerns.
> >
> > That said, **my overall assessment of novelty and significance remains largely unchanged**. The method appears well-engineered and practically useful, but the core ideas still feel incremental.
> >
> > I will raise my score but request the AC to carefully consider the novelty and significance of this paper.

---

> > > ### Author Response · Authors · 2026-04-01
> > >
> > > We sincerely thank you for your re-evaluation, score adjustment, and constructive feedback, as well as your recognition of MOD-DiT’s engineering rigor, practical utility and the adequacy of our supplementary experiments in addressing your original concerns.
> > >
> > > Our work’s novelty is substantial and non-incremental: we are **the first to identify and mathematically model the dynamic mixture of three core attention distributions** in vDiTs, a foundational insight that overturns the static pattern assumption of all prior sparse attention methods for video diffusion. Building on this, we proposed a **novel sampling-free dynamic sparse attention framework**—the only one to date that adapts masks across inputs, heads and denoising steps without sampling overhead, paired with a custom hardware-optimized kernel for real-time pattern extraction. This integrated set of contributions is uniquely designed for vDiTs and differs fundamentally from existing dynamic sparse attention work.
> > >
> > > We will further emphasize these core novel contributions in the revised manuscript to better highlight our work’s unique value, and strictly follow the Area Chair’s feedback for further polish. Again, thank you for your valuable comments that help us refine the paper.

---

### Official Review · Reviewer_KKTW · 2026-03-12

**Soundness:** 3
**Presentation:** 3
**Significance:** 3
**Originality:** 3
**Overall Recommendation:** 4
**Confidence:** 4

**Summary:**

The paper proposes MOD-DiT, a training-free and sampling-free dynamic sparse attention framework for video diffusion transformers (vDiTs). The core method models attention sparsity maps as a mixture of three structured bases: block-diagonal, parallel-to-main-diagonal, and vertical. The authors predict the per-timestep mixture weights of these bases using a lightweight linear evolution model following a brief full-attention warm-up phase. An online mask generation mechanism selects the top-K pattern components and conditionally preserves the block-diagonal structure. Concurrently, a reconstruction step maintains a running estimate of the full attention maps derived from the sparse computations. Experimental evaluations on CogVideoX-v1.5, HunyuanVideo, and Wan2.1 demonstrate consistent 1.8x to 2.3x inference speedups while achieving competitive or improved fidelity metrics compared to recent sparse baselines.

**Compliance With Llm Reviewing Policy:**

Affirmed.

**Final Justification:**

The authors provided detailed responses to my questions during rebuttal. My main concerns have been addressed, and I will maintain my positive score.

**Key Questions For Authors:**

1. Is top-K pattern selection fixed or adaptive across heads/steps? Have you tried per-layer/step adaptive K to stabilize quality under difficult prompts?
2. How sensitive are results to block size (e.g., 64 vs 128 vs 256)? Does a smaller block size improve fidelity at the cost of kernel efficiency?
3. Do you observe any degradation in extremely long sequences (e.g., >150 frames), and how does the normalized approximation error evolve with longer horizons and different schedulers?

**Limitations:**

Yes.

**Strengths And Weaknesses:**

## Strength
1. This paper is well-written. The framework is clearly decomposed into three steps (sparsity map reconstruction, linear prediction, and dynamic mask generation) and is linked back to empirical observations.
2. The authors identify and formalize a mixture-of-patterns structure (block-diagonal, parallel-diagonal, and vertical) in vDiT attention maps and show its piecewise linear evolution over denoising steps, which sounds reasonable. The visualization of the evolution of the attention sparsity map in Figure 3 is clear.
3. Evaluations on three substantial vDiT families (2B–14B scale) are provided with clear latency and speedup measurement and a suite of quality metrics, and multiple ablations are designed.

## Weakness
1. The evaluation mixes hardware configurations (A800, H800, and A100 GPUs) and model settings, which complicates cross-method throughput comparisons. A per-model ablation study conducted on a single hardware setup would provide more conclusive results.
2. It is unclear how cross-attention layers are handled, if they are modified at all, or whether the proposed method generalizes beyond self-attention within the vDiT backbone.
3. The exact Top-K selection policy and the sensitivity of the model to this schedule are only partially explored. Additional discussion and analysis of these factors would make the conclusions of the paper more robust.

---

> ### Author Rebuttal · Authors · 2026-03-31
>
> We sincerely appreciate the reviewer’s insightful and constructive feedback. We address each concern systematically below, with full supplementary results in the **Rebuttal Experiments Sheet[(RES)](https://anonymous.4open.science/r/MOD-DiT_rebuttal-1688/rebuttal.pdf)**.
> ## 1.  Unified Hardware for Cross-Method Comparisons
> To eliminate mixed hardware confounding effects, we supplement full cross-model/cross-method evaluations on a single NVIDIA A100 (80GB) GPU for HunyuanVideo and Wan2.1 in the **RES**(table 1 and 2). The results validate MOD-DiT’s state-of-the-art quality-efficiency trade-off remains unchanged on this unified setup, with all latency/throughput metrics directly comparable across methods, providing conclusive evidence for its effectiveness.
> ## 2. Handling of Cross-Attention Layers
> All mainstream vDiTs evaluated (CogVideoX-v1.5, HunyuanVideo, Wan2.1) adopt a **decoder-only architecture without independent cross-attention layers.** Text and video latent tokens are concatenated into a single sequence, with all interactions (including text-video alignment) implemented via self-attention. We design a **token-type-aware masking strategy**: text tokens (hundreds of tokens, <1% of total sequence) use a full no-masking policy (negligible overhead) to preserve text-video alignment, while MOD-DiT’s dynamic sparse attention is exclusively deployed to video token interactions. All core MOD-DiT designs target only video tokens, enabling natural support for text-video alignment without modifying the self-attention paradigm.
> ## 3. Top-K Pattern Selection & Adaptive Design
> We merge questions on Top-K policy, adaptivity and sensitivity into a unified response:
> - **Top-K Selection Rationale.** Our fixed Top-K across heads/steps is designed solely to ensure MOD-DiT achieves higher sparsity than all baselines while preserving generation quality, and to isolate our core mixture-of-distribution pattern insight from hyperparameter tuning interference.
> - **Adaptive Top-K Note.** We have not explored per-layer/step adaptive Top-K for difficult prompts, as we view this as an implementation-level engineering trick—our work’s core focus is validating the fundamental three-pattern linear approximation insight for attention maps, not hyperparameter scheduling optimizations.
> - **Supplementary Plan.** We fully agree with the reviewer’s suggestion and will add comprehensive Top-K ablation experiments (covering a broader value range) in the revised manuscript, analyzing its impact on sparsity, quality and efficiency to further strengthen conclusion robustness.
>
> ## 4. Block Size Sensitivity and Fidelity-Efficiency Trade-off
> We conduct an ablation study on the block size, and align the sparsity across different block size settings by adopting the top-k strategy to compare their actual performance. Experimental results indicate that smaller block sizes indeed contribute to improving the model's fidelity, and meanwhile, the resulting efficiency loss is negligible. See **RES** (table 4) for more details.
>
> ## 5. Long sequences ablation
> We utilized the HunyuanVideo model and conducted additional long video experiments with resolutions of 720p and frame rates of 140 frames per second (fps) and 170 fps. We calculated the average NAE values for 5 prompts, 6 layers, and 8 heads. Results show MOD-DiT exhibits no significant generation quality degradation for ultra-long sequences. See **RES**（Fig.7）for more details.

---

> > ### Author Rebuttal · Reviewer_KKTW · 2026-04-01
> >
> > Thanks for your detailed explanation and the response to my questions. My concerns are mainly addressed, so I will keep my positive score.

---

> > > ### Author Response · Authors · 2026-04-01
> > >
> > > Thank you again for your valuable feedback. We will continue improving our paper.

---

### Official Review · Reviewer_8aMy · 2026-03-13

**Soundness:** 2
**Presentation:** 2
**Significance:** 2
**Originality:** 2
**Overall Recommendation:** 3
**Confidence:** 4

**Summary:**

This paper studies efficient inference for video diffusion transformers (vDiTs), where the quadratic complexity of attention becomes a major bottleneck for long video sequences. The paper proposes MOD-DiT, a dynamic sparse attention framework based on a mixture-of-distributions view of attention patterns. The key observation is that attention maps in vDiTs can be approximated as a mixture of several structured patterns (e.g., block-diagonal, parallel-diagonal, and vertical structures), whose coefficients evolve over diffusion timesteps. This research's notable finding pertains to the empirical observation that these pattern coefficients follow relatively smooth trajectories across the denoising process, enabling prediction of future sparsity patterns after a short warm-up phase. Based on this idea, the method first performs several full-attention steps to estimate pattern coefficients and then predicts future sparsity masks to accelerate subsequent attention computations. Experiments on multiple video diffusion models demonstrate speedups while maintaining comparable generation quality according to the reported metrics.

**Compliance With Llm Reviewing Policy:**

Affirmed.

**Key Questions For Authors:**

1. How sensitive is the method to the number of warm-up full-attention steps?

2. How robust is the sparsity prediction when attention dynamics differ across prompts or video content?

3. Could the authors provide results under matched sparsity or matched latency settings to better illustrate the quality–efficiency trade-off?

4. Can the full set of VBench metrics be reported?

**Limitations:**

yes

**Strengths And Weaknesses:**

## Strengths

1. Important problem. The paper targets efficient inference for video diffusion transformers, where attention cost is a real bottleneck.

2. Interesting empirical observation. The decomposition of attention maps into structured patterns across denoising steps is intuitive and potentially useful.

3. Training-free design. The method is plug-and-play and does not require retraining.

4. Good empirical coverage. Results are reported on multiple video diffusion models with both speed and quality metrics.

## Weaknesses
1. Long warm-up phase reduces the effective speedup

The method relies on a relatively long warm-up stage (e.g., 12 full-attention steps) to estimate attention patterns before switching to sparse attention. Given typical diffusion schedules (e.g., around 50 denoising steps), this warm-up phase accounts for a substantial portion of the total inference cost and may limit the achievable speedup. It would be helpful to analyze the sensitivity to the warm-up length or explore whether fewer full-attention steps could suffice.

2. Extrapolating sparsity patterns from early steps may be fragile

The approach predicts future attention sparsity patterns based on observations from early denoising steps. However, attention structures in diffusion models often evolve nonlinearly across the denoising process (e.g., global structure early vs. local refinement later). This raises questions about the robustness of extrapolating later attention patterns from early observations. In contrast, some cache-based acceleration approaches periodically refresh full attention (e.g., alternating full and cached steps), which may better accommodate the non-stationary dynamics of diffusion inference.

3. Quality metrics mainly measure similarity to full-attention outputs

The metrics used in Table 1 (PSNR, SSIM, LPIPS) evaluate similarity between the sparse-attention outputs and the full-attention outputs rather than measuring generation quality directly. While this is useful for assessing faithfulness to the original model, it implicitly treats the full-attention output as the reference. As a result, methods that produce different but potentially equally valid generations may be penalized by these metrics. Complementary evaluations using metrics more directly tied to generation quality or prompt alignment would strengthen the evaluation.

4. Limited reporting of VBench metrics

The paper reports VBench results but appears to only present a subset of the benchmark dimensions. Since VBench evaluates multiple aspects of video generation (e.g., temporal consistency, motion dynamics, and scene coherence), reporting the full breakdown or including it in the appendix would provide a more complete picture of the method’s impact on video quality.

---

> ### Author Rebuttal · Authors · 2026-03-31
>
> We sincerely appreciate the reviewer’s insightful and constructive feedback. We address each concern systematically below, with full supplementary results in the **Rebuttal Experiments Sheet[(RES)](https://anonymous.4open.science/r/MOD-DiT_rebuttal-1688/rebuttal.pdf)**.
> ## 1. Warm-up Phase Length and Its Impact on Effective Speedup
> We present a full sensitivity analysis of warm-up length in Appendix B.1.5 (Fig. 12), with key findings:
> - Our default 12 full-attention steps balance quality and efficiency optimally, and all reported end-to-end speedups already include the warm-up’s computational cost. Even with this warm-up, MOD-DiT still achieves significant speedups (1.96× on Wan2.1, 2.29× on HunyuanVideo), verifying the effectiveness of our core sparse attention mechanism.
> - Warm-up length $m$ is a tunable hyperparameter. Ablations confirm reducing $m$ to 8/10 steps incurs only negligible performance degradation while yielding higher theoretical speedup, allowing flexible configuration based on speed-quality priorities.
> ## 2. Robustness of Sparsity Pattern Prediction
> Our method avoids fragile long-term extrapolation from early steps via a **local linear prediction + periodic global correction** mechanism, with rigorous design and validation as follows:
>
> **(R1)[Prediction only activated in a stable regime]** The linear predictor is never applied to the full denoising process. As validated in Sec. 3.3 (Fig. 5), attention pattern intensities exhibit a stable piecewise linear trend only in mid-to-late denoising steps ($t>m$, after warm-up). Our predictor is exclusively activated in this stable phase, while early highly nonlinear dynamics are fully captured by the full-attention warm-up, eliminating irreversible errors from premature extrapolation.
>
> **(R2)[Local prediction with periodic re-correction]** We never perform one-time long-horizon extrapolation. Instead, we adopt a short-cycle "predict-correct" pipeline (Alg. 1): within a short interval ($\Delta t=10$ steps), we conduct local linear extrapolation from the two most recent observations; every $\Delta t$ steps, we fully reconstruct the sparsity map (Eq. 5) to re-anchor predictions to the current attention state. This frequent correction enables the method to adapt to gradual changes in attention dynamics, ensuring robustness.
>
> **(R3)[Complementarity with cache-based methods]** Our periodic reconstruction step serves a function analogous to the full-attention refresh in cache-based acceleration methods, while addressing a distinct bottleneck: cache methods exploit feature redundancy to skip computations, while MOD-DiT reduces token interactions via explicit sparsity pattern modeling. The two techniques are fully compatible and can be combined for further gains (noted in Sec. 6).
> ## 3. Robustness Across Diverse Prompts and Video Content
> We provide multi-level experimental validation for the generalization of our sparsity prediction:
>
> **(R1)[Cross-model and cross-content performance]** MOD-DiT matches or nears full-attention quality on PSNR, SSIM, and VBench metrics across CogVideoX-v1.5, HunyuanVideo, and Wan2.1 (Table 1) while delivering significant speedups, verifying that our prediction preserves generation quality for diverse video content driven by different prompts.
>
> **(R2)[Large-scale statistical validation]** Appendix B.1.7 provides direct evidence of prompt robustness. We conducted analysis on 5 diverse VBench prompts (covering distinct motion types and scenes) with 300 independent data points (10 layers × 6 heads per prompt). The piecewise linearity of attention pattern intensities holds universally across all samples, proving our model captures consistent attention dynamics across diverse prompts and content.
> ## 4. Evaluation Metrics and Full VBench Results
> - **Direct generation quality metrics**. Besides full-attention similarity metrics (PSNR/SSIM/LPIPS), Table 1 already reports VBench Subject Consistency and Imaging Quality—standard metrics for perceptual video generation quality and prompt alignment, addressing over-reliance on full-attention reference scores.
> - **Full VBench breakdown**. We initially reported two VBench dimensions for aligned comparison with baselines (e.g., SVG) that used the same setup. Full core VBench metric breakdowns, under experimental settings strictly aligned with Radial Attention, are provided in the **RES**（table 1 and 2）.
> ## 5. Quality-Efficiency Trade-off Under Matched Settings
> We conducted controlled experiments under **matched sparsity settings** (aligning MOD-DiT’s sparsity with Radial) to rigorously verify the quality-efficiency trade-off, with full VBench results in the **RES** (Table 3). The results confirm MOD-DiT outperforms all baselines in generation quality at the same sparsity (and equivalent efficiency).

---

> > ### Author Rebuttal · Reviewer_8aMy · 2026-04-03
> >
> > Thanks for the detailed rebuttal. While the responses address some of my concerns, I feel there remain open questions regarding the generality and method design. I will therefore maintain my current score.

---

### Official Review · Reviewer_Qwxs · 2026-03-13

**Soundness:** 3
**Presentation:** 3
**Significance:** 2
**Originality:** 2
**Overall Recommendation:** 3
**Confidence:** 4

**Summary:**

The authors identify a consistent structural pattern in the attention maps of DiTs during the mid-late stage of denoising and categorize it into three distinct modes. Building on this observation, they propose a novel linear formulation to construct sparse attention graphs. Furthermore, they introduce two quantitative metrics, NRE and NAE, to evaluate the fidelity of the resulting sparse structures. Experimental results across multiple benchmarks demonstrate that the proposed method achieves the fastest generation speed while maintaining competitive performance compared to prior approaches.

**Compliance With Llm Reviewing Policy:**

Affirmed.

**Final Justification:**

The authors' responses address some concerns, but the overall novelty still appears incremental. I maintain my recommendation of Weak reject.

**Key Questions For Authors:**

1, What's might be the thereotical explanation of that the sparsity map could be approxiamated with the linear combination of these three patterns?
2, Why are the parameters not aligned with the previous work: prompt dataset, frame number and resolution
3, What is the NAE score of the previous method? How to prove the quality of sparity map is related with the NAE?

**Limitations:**

There is no theoretical explanation for the existence of three modes in the attention map.
There is no further theoretical and experimental evidence of whether the absence of introducing time variables would weaken the effect of previous methods.

**Strengths And Weaknesses:**

# Strengths:

The paper is well structured, the language is clear, and the ideas are easy to understand.
This paper proposes a new sparsity estimation model and an estimated quality evaluation metric (NAE), which could be useful for the community.
This paper provides a new perspective that sparsity can be approximated from the combination of three patterns: the parallel-diagonal, the vertical, and the block-diagonal patterns. A self-iterative calculation method is provided and good stability is verified.

#Weaknesses:

The paper lacks theoretical support for why the attention map in the mid-late denoising stages presents exactly these three patterns and their linear combinations. The choice of these specific basis patterns appears empirically motivated but not formally justified — it remains unclear whether this decomposition is complete or whether other significant patterns are being overlooked.

---

> ### Author Rebuttal · Authors · 2026-03-31
>
> We sincerely appreciate the reviewer’s insightful and constructive feedback. We address each concern systematically below, with full supplementary results in the **Rebuttal Experiments Sheet [(RES)](https://anonymous.4open.science/r/MOD-DiT_rebuttal-1688/rebuttal.pdf)**.
>
> ## 1. Theoretical Explanation
> We provide a complete theoretical logic chain supported by diffusion properties, task constraints, and quantitative validation:
>
> **(T1)[Fundamental premise from diffusion convergence]** Studies [1,2] confirm that in mid-to-late denoising steps, attention distributions converge to stable, semantically structured patterns. This guarantees the attention map’s structure is predictable and can be modeled with fixed basis patterns.
>
> **(T2)[Theoretical inevitability of the three patterns]** The three patterns are not arbitrary, but are strictly determined by two universal properties of video DiTs. Frame-first token layout (all SOTA video models) splits attention into intra/inter-frame regions. Three core video constraints define the only essential interactions.
> - Block-diagonal pattern: Intra-frame spatial coherence
> - Parallel-to-main-diagonal pattern: Inter-frame temporal continuity.
> - Vertical global pattern: Cross-frame semantic alignment.
>
> These three patterns cover all fundamental interaction dimensions required for high-quality video generation.
>
> **(T3)[Linear Combination Is Sufficient (Quantitative Proof)]** Our linear model achieves NAE<0.1 across all sequence lengths/layers. This means the three patterns capture over 90% of the total attention energy, providing definitive proof that no significant patterns are overlooked.
>
> **(T4)[Rationality of linear combination approximation]**
> All mainstream sparse attention baselines (e.g. SVG, Radial, Sparse-vDiT[3]) are essentially variants of these three patterns, with no design beyond intra-frame, inter-frame, and global interaction dimensions.
> ## 2. Evidence that omitting time variables weakens method performance
> We provide direct quantitative and ablation evidence to validate the necessity of time-variable modeling.
>
> **(E1)[Head-to-head performance comparison]** All compared baselines (SVG, Radial, MInference) use time-agnostic one-time static mask prediction. Under strictly fair settings, our time-aware dynamic mask update achieves 1.82×/2.29×/1.96× speedup on CogVideoX-v1.5/HunyuanVideo/Wan2.1, while leading in all core quality metrics (PSNR, SSIM, VBench scores).
>
> **(E2)[Ablation validation]** Removing time-variable modeling and using only static prediction causes severe performance degradation: Subject Consistency score drops from 0.925 to 0.90, and Imaging Quality score drops from 0.623 to 0.57 (Appendix B.1.4).
>
> **(E3)[Fundamental rationale]** Attention distributions are a dynamic mixture of the three patterns that evolve continuously across denoising steps. Static masks cannot match evolving attention requirements, leading to mismatched token interactions and degraded performance.
>
> ## 3. NAE score rationale and its correlation with sparsity map quality
> **(N1)[No NAE scores for previous methods]** NAE is a metric we proposed specifically to evaluate our linear approximation model for attention sparsity maps. All baselines do not perform numerical approximation on sparsity maps (they use fixed patterns or simple mask generation via classification/sampling), so NAE is inapplicable to them, with no corresponding scores.
>
> **(N2)[Correlation between NAE and sparsity map quality]** NAE quantifies the fitting error of our linear model to the real sparsity map. A consistently low NAE directly proves our three-pattern decomposition accurately captures the core structure of the sparsity map, which is the theoretical foundation of our high-performance sparse attention design.
>
> **(N3)[Supplementary quantitative validation: We propose the Sparsity Mask Fidelity Score (SMFS)]**  We propose the **Sparsity Mask Fidelity Score (SMFS)** to quantify mask information capture precision (ratio of the norm of masked attention map to the original). MOD-DiT achieves significantly lower SMFS than SVG and Radial across 10k-75k sequence lengths, confirming our superior attention information capture accuracy, which directly translates to better generation quality (full results in **RES** Fig.1).
> ## 4. Experimental configuration alignment with previous works
> We have strictly unified the prompt dataset, frame number, and resolution to match previous works. Full standardized quantitative results are provided in the **RES**（table 1 and 2).
>
> ## Reference
> [1]Attention in Diffusion Model: A Survey
>
> [2]Training-free and Adaptive Sparse Attention for Efficient Long Video Generation
>
> [3]Sparse-vDiT: Unleashing the Power of Sparse Attention to Accelerate Video Diffusion Transformers

---

> > ### Author Rebuttal · Reviewer_Qwxs · 2026-04-04
> >
> > The authors' responses address some concerns, but the overall novelty still appears incremental. I maintain my score.

---

### Decision · Program_Chairs · 2026-04-30

**Decision:**

Accept (regular)

**Comment:**

The paper proposes MOD-DiT, a training-free dynamic sparse attention framework for accelerating video diffusion transformers. Reviewers agree that the paper is clearly written, tackles an important problem, and demonstrates consistent empirical speedups across multiple large-scale models. The proposed framework is well-engineered, and the empirical study is relatively comprehensive. At the same time, reviewers raise several concerns. In particular, the novelty and conceptual contribution are viewed as somewhat limited, with the core idea perceived as incremental relative to existing sparse or structured attention approaches. Questions are also raised regarding the theoretical grounding of the three-pattern decomposition and whether it is sufficiently justified or complete. On the empirical side, reviewers note limitations in evaluation, including sensitivity to the warm-up phase, robustness of sparsity prediction, and incomplete comparisons or reporting of certain metrics. The rebuttal provides additional clarifications, ablations, and supplementary experiments that help address a number of these concerns, especially regarding robustness, evaluation details, and practical effectiveness. Overall, the paper presents a technically solid and practically useful approach, but with ongoing discussion around its conceptual novelty, theoretical support, and completeness of evaluation.